# Concerted action of the MutLβ heterodimer and Mer3 helicase regulates the global extent of meiotic gene conversion

Yann Duroc[1,2†], Rajeev Kumar[1,2†‡], Lepakshi Ranjha[3†§], Céline Adam[1,2], Raphaël Guérois[4,5], Khan Md Muntaz[6], Marie-Claude Marsolier-Kergoat[4,5,7], Florent Dingli[8], Raphaëlle Laureau[1,2], Damarys Loew[8], Bertrand Llorente[6], Jean-Baptiste Charbonnier[4,5], Petr Cejka[3*§], Valérie Borde[1,2*]

[1]Institut Curie, PSL Research University, CNRS UMR3664, Paris, France; [2]Université Pierre et Marie Curie, Paris, France; [3]Institute of Molecular Cancer Research, University of Zurich, Zurich, Switzerland; [4]I2BC, iBiTec-S, CEA, CNRS UMR 9198, Université Paris-Sud, Gif-sur-Yvette, France; [5]Université′ Paris Sud, Orsay, France; [6]CRCM, Inserm U1068, Institut Paoli-Calmettes, Aix-Marseille Université UM105, CNRS UMR7258, Marseille, France; [7]Musée de l'Homme, CNRS UMR 7206, Paris, France; [8]Institut Curie, Centre de Recherche, PSL Research University, LSMP, Paris, France

*For correspondence: cejka@ imcr.uzh.ch (PC); valerie.borde@ curie.fr (VB)

[†]These authors contributed equally to this work

Present address: [‡]Institut Jean-Pierre Bourgin, UMR1318, INRA-AgroParisTech, Paris, France; [§]Institute for Research in Biomedicine, Università della Svizzera italiana, Bellinzona, Switzerland

Competing interests: The authors declare that no competing interests exist.

**Abstract** Gene conversions resulting from meiotic recombination are critical in shaping genome diversification and evolution. How the extent of gene conversions is regulated is unknown. Here we show that the budding yeast mismatch repair related MutLβ complex, Mlh1-Mlh2, specifically interacts with the conserved meiotic Mer3 helicase, which recruits it to recombination hotspots, independently of mismatch recognition. This recruitment is essential to limit gene conversion tract lengths genome-wide, without affecting crossover formation. Contrary to expectations, Mer3 helicase activity, proposed to extend the displacement loop (D-loop) recombination intermediate, does not influence the length of gene conversion events, revealing non-catalytical roles of Mer3. In addition, both purified Mer3 and MutLβ preferentially recognize D-loops, providing a mechanism for limiting gene conversion in vivo. These findings show that MutLβ is an integral part of a new regulatory step of meiotic recombination, which has implications to prevent rapid allele fixation and hotspot erosion in populations.

## Introduction

Meiotic recombination is a key evolutionary process that promotes genome diversification in sexually reproducing organisms. Recombination shuffles parental genomes through genetic exchanges leading to crossovers (COs) or noncrossovers (NCOs). During meiotic prophase, repair of Spo11-induced DNA double-strand breaks (DSBs) by recombination into COs is crucial for the formation of gametes with normal chromosome content. Indeed, COs ensure a physical link between homologs that allows them to properly segregate to opposite poles during meiosis I division (*Hunter, 2015*). DSBs occur in excess over COs, so many DSBs repaired as noncrossovers (NCOs), without exchange of flanking sequences.

Both COs and NCOs involve the formation of heteroduplex DNA, which can yield gene conversions after mismatch repair (*Figure 1*). Gene conversions consist of the unidirectional transfer of

**Figure 1.** Effect of gene conversion during meiotic recombination on allele shuffling and erosion of cis-acting hotspot sequences. Two homologous chromatids are shown in red and blue, the red one having a DSB formed by Spo11. A, B and C represent alleles, with the b allele being present on the red homolog. The star represents a cis-acting hotspot promoting sequence. Following DNA end resection and strand invasion of the intact DNA duplex, the red homolog sequences are copied from the blue one, creating heteroduplexes (indicated by a green square), that are next corrected by mismatch repair, leading either to gene conversion or restoration. After gene conversion, the b allele on the broken chromosome has been converted to the B allele, and the hotspot promoting sequence has been converted to the blue sequence.

genetic information from one parental chromosome to the other. Gene conversions therefore result in non-Mendelian segregation and transmission distortion, and also largely contribute to the extinction of recombination-promoting sequences (hotspots) (reviewed in *Cole et al., 2012*). Despite the importance of gene conversion in promoting genetic diversity, the factors that regulate the extent of sequences converted at each recombination site are not known.

To repair a subset of DSBs into COs, cells mainly employ a meiosis-specific pathway that ensures an even distribution of COs across the genome (*Zhang et al., 2014*). For this, after the initial strand invasion of a homologous DNA template by the 3' end of the resected DSB, the resulting D-loop intermediate is stabilized by a group of proteins called 'ZMM', that lead to the single-end invasion (SEI) intermediate (*Börner et al., 2004*; *Hunter and Kleckner, 2001*). The ZMM proteins in budding yeast comprise Zip1, Zip2, Zip3, Zip4, Spo16, the Mer3 helicase, and the Msh4-Msh5 heterodimer (*Lynn et al., 2007*; *Shinohara et al., 2008*). A proposed function of the ZMM proteins is to protect recombination intermediates from the action of helicases, such as Sgs1 in budding yeast (*Jessop et al., 2006*; *Oh et al., 2007*), which may dismantle the joint molecules, leading to synthesis-dependent strand annealing (SDSA) resulting in non-crossovers (*Figure 1*). These ZMM- stabilized intermediates then mature in double Holliday junctions, which are specifically resolved into COs at the pachytene stage, likely by MutLγ (*Allers and Lichten, 2001*; *Hunter and Kleckner, 2001*; *Zakharyevich et al., 2012*).

MutLγ (Mlh1-Mlh3), a heterodimer related to the bacterial mismatch repair MutL complex, forms foci on pachytene mouse, human or plant chromosomes in numbers that correspond to CO numbers. In the absence of MutLγ, CO frequency is reduced in mouse meiosis (*Baker et al., 1996*; *Edelmann et al., 1996*; *Guillon et al., 2005*; *Lipkin et al., 2002*) as well as in budding yeast (*Hunter and Borts, 1997*; *Nishant et al., 2008*). CO remaining in the absence of MutLγ likely result from cleavage of recombination intermediates by alternate structure-specific nucleases (Mus81, Yen1, Slx1-4) that normally play a minor role and are only activated late in the cell cycle (*De Muyt et al., 2012*; *Matos et al., 2011*; *Zakharyevich et al., 2012*). The integrity of the nuclease active site of Mlh3 is required for MutLγ function in CO formation, and MutLγ has thus been proposed to cleave and resolve double Holliday junctions into crossovers (*Nishant et al., 2008*; *Ranjha et al., 2014*; *Rogacheva et al., 2014*; *Zakharyevich et al., 2012*).

In mismatch repair (MMR), the MutS homologs (Msh2-Msh3 and Msh2-Msh6) are involved in mismatch recognition and recruit either of two MutL heterodimers, MutLα (Mlh1-Pms1 in yeast, MLH1-PMS2 in human) and to a lesser degree MutLγ (Mlh1-Mlh3) (*Li and Modrich, 1995*; *Lipkin et al., 2000*; *Räschle et al., 1999*; *Wang et al., 1999*). Both MutLα and to a lesser degree MutLγ bear an endonuclease activity that is involved in MMR (*Gueneau et al., 2013*; *Kadyrov et al., 2006*). MMR corrects bases misincorporated during replication, but also mismatches formed during homologous recombination (*Kadyrov et al., 2006*, *2007*; *Wang et al., 1999*).

The third MutL heterodimer, Mlh1-Mlh2 (MLH1-PMS1 in mammals) or MutLβ, has an elusive function. No biochemical activity has been described, and the human MutLβ has no MMR activity in in vitro complementation assays (*Räschle et al., 1999*). Consistently, in mammalian cells, PMS1 deficiency leads to a very small increase in mutation frequency (*Prolla et al., 1998*). Yeast *mlh2Δ* cells show no increase in mutation rates, except a slight defect in the repair of a subset of frameshift mutations (*Campbell et al., 2014*; *Harfe et al., 2000*). In addition, Mlh2 forms spontaneous Msh2-dependent foci in S-phase that partially co-localize with Pms1, suggesting that MutLβ may be a non-essential accessory factor of MutLα (*Campbell et al., 2014*). PMS1-/- mice are fertile, suggesting no gross defect in meiotic recombination, contrary to MLH1-/- or MLH3-/- (*Prolla et al., 1998*). In yeast meiosis, *mlh2Δ* cells show normal spore viability. However, an increase in conversion of markers flanking several recombination hotspots was observed in *mlh2Δ* cells (*Abdullah et al., 2004*; *Wang et al., 1999*).

Here, we used a proteomic approach to identify meiotic partners of Mlh1 in budding yeast. We discovered that MutLβ controls a new regulatory step of meiotic recombination that limits the length of meiotic gene conversion tracts associated with COs and NCOs. This regulation requires a physical interaction between the MutLβ complex and the meiosis-specific helicase, Mer3. This work opens new perspectives to investigate mechanisms that control the genetic diversity created by meiotic recombination.

# Results

## Obtaining a functional tagged allele of Mlh1

The functionality of Mlh1 is affected when tagged at its N- or C-terminus, which has been a major obstacle for in vivo molecular studies. To obtain a functional tagged allele of Mlh1, we used the crystal structure of the yeast Mlh1-Pms1 C-terminal domain that we previously reported (*Gueneau et al., 2013*), and chose a solvent accessible loop in the Mlh1 C-terminal domain remote from the endonuclease site and from the Exo1 binding site to insert internal tags and perform molecular studies of Mlh1 complexes during meiotic recombination (*Figure 2a*). The resulting Flag- and HA- *MLH1* tagged alleles were fully functional in mismatch repair (*Figure 2b*). In addition, in meiosis, both alleles conferred full spore viability, contrary to *mlh1Δ* (*Figure 2c*). Finally, both His-Flag- and HA- internally tagged *MLH1* alleles produced wild-type frequencies of crossing over at the

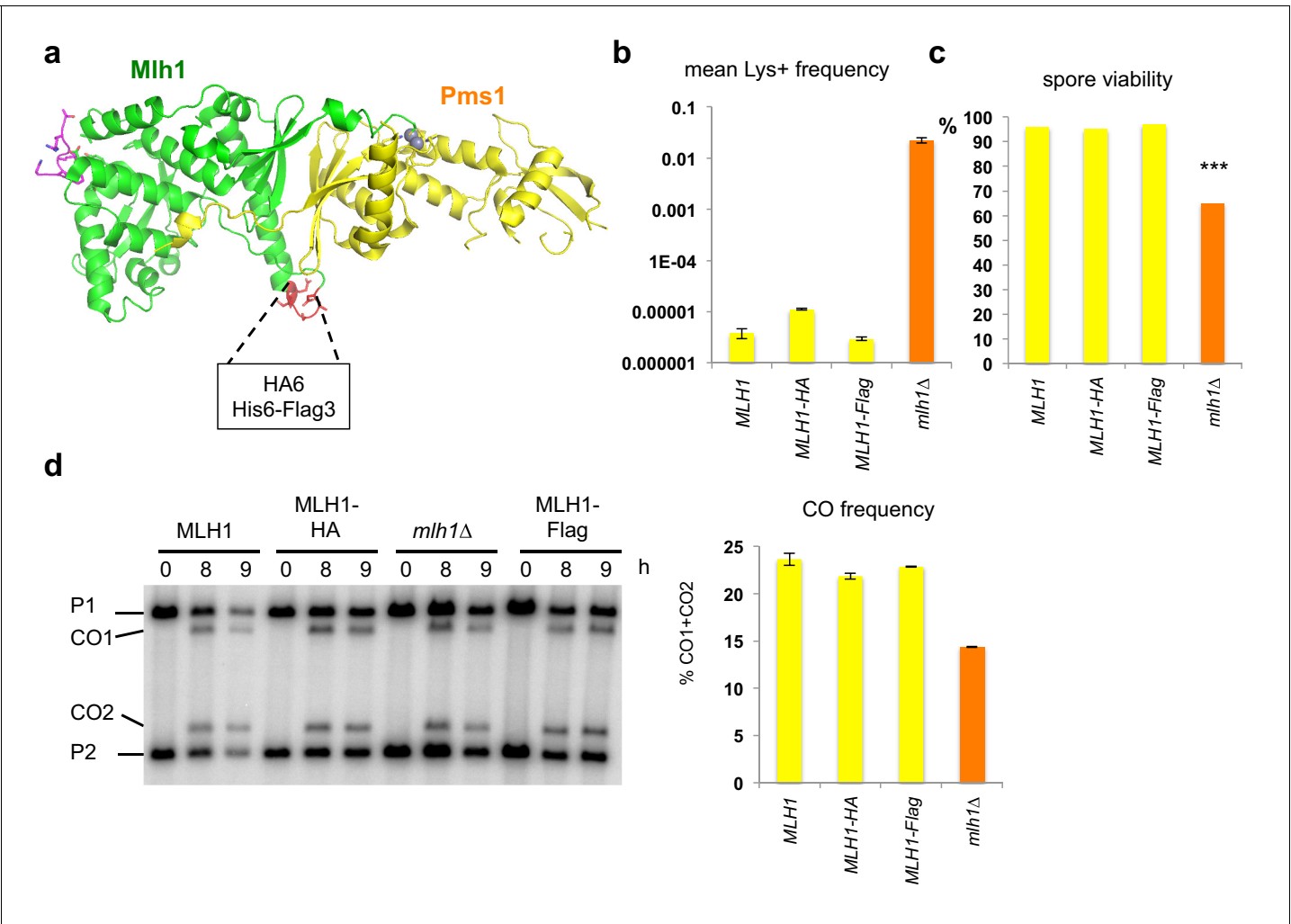

**Figure 2.** Mlh1 alleles tagged internally in the Cter domain are fully functional for MMR and meiotic recombination. (a) Crystal structure of the C-terminal region of *S. cerevisiae* Mlh1-Pms1 heterodimer (pdb code 4FM0) (*Gueneau et al., 2013*). The Mlh1 and Pms1 regions are colored in green and yellow, respectively. The two metal ions of the endonuclease site are represented by grey spheres. The peptide containing the Mlh1-binding motif for Exo1 is colored in magenta. (b) Mutator assay. Frequency of reversion to Lys+ in cells containing the indicated MLH1 genotype at its endogenous locus. Values are the mean of 9 independent colonies ± SEM. (c) Spore viability of diploid SK1 strains bearing the indicated MLH1 genotype at its endogenous locus. *MLH1*: VBD1311 (188 tetrads), Mlh1-HA: VBD1456 (22 tetrads), Mlh1-Flag: VBD1337 (26 tetrads), *mlh1Δ*: VBD1494 (178 tetrads). ***p$<5.10^{-5}$, Fisher's exact test. (d) Crossing over frequency at the *HIS4LEU2* hotspot monitored by Southern blot at the indicated times in meiosis. Positions of parental bands (P1 and P2) and of the recombinant crossover products (CO1 and CO2) are indicated. Graph shows quantification at 9 hr from two independent biological replicates ± SEM. Same strains as in (c).

*HIS4LEU2* meiotic recombination hotspot (*Figure 2d*). We conclude that our internally tagged *MLH1* alleles are fully functional and suitable to perform molecular studies of Mlh1 complexes during meiotic recombination.

## Mlh1 interacts with the Mer3 helicase

To identify Mlh1 partners in meiosis, we performed a Flag pulldown of the Mlh1 complexes in synchronous meiotic cells at 4 hr after induction of meiosis, the expected time of meiotic recombination. As anticipated, we recovered Pms1, Mlh2 and Mlh3, the three known MutL partners of Mlh1. In addition, we recovered the Mer3 helicase of the procrossover ZMM group of proteins (*Figure 3a*). We confirmed that Mlh1 and Mer3 reciprocally co-immunoprecipitate in meiotic cells using doubly tagged Mlh1-HA Mer3-Flag cells (*Figure 3b*).

Moreover, the budding yeast Mer3 and Mlh1 as well as their respective mouse orthologs HFM1 and MLH1 interacted with each other in yeast two hybrid assays, showing that this interaction is evolutionarily conserved (*Figure 3c* and *Figure 3—figure supplement 1a,b*). In both species, the C-terminal part of Mer3 was essential and sufficient for the two-hybrid interaction with Mlh1 (*Figure 3c* and *Figure 3—figure supplement 1a,b*). However, in meiotic cells, Mlh1 interaction was still observed with Mer3 deleted of its C-terminal part (*Figure 3c and d*, Mer3ΔC). Instead, the Mer3 protein deleted for its IG-like domain (Mer3ΔIG) no longer interacted with Mlh1 in meiotic cells, suggesting that a third protein interacting with this region may bridge the in vivo interaction (*Figure 3c* and *Figure 3—figure supplement 1c*).

## Mer3 interacts with the MutLβ (Mlh1-Mlh2) complex both in vitro and in meiotic cells

To identify the factor that may help bridge the interaction between Mer3 and Mlh1 in vivo, we performed a Mer3-Flag pulldown from synchronous meiotic cells. Mer3 not only pulled down Mlh1, but also Mlh2, the Mlh1 partner in the MutLβ heterodimer (*Figure 3—figure supplement 1d*). We found that Mlh2 was important to mediate the interaction between Mlh1 and Mer3 in vivo, indicating formation of a tripartite complex between Mer3 and Mlh1-Mlh2 (MutLβ, *Figure 3d*).

To extend this observation, we further tested for direct interactions using purified proteins. Recombinant Mer3 strongly interacted in vitro with the purified MutLβ complex, showing that the interaction is direct (*Figure 4a*). Mer3 also interacted with the component Mlh2 protein and also with Mlh1 to a lesser degree (*Figure 4b* and *Figure 4—figure supplement 1*). In accord, a direct interaction between Mer3 and Mlh1 could be observed in a yeast two hybrid assay, independent of the presence of Mlh2 (*Figure 4—figure supplement 2*). However, this interaction does not seem to be sufficient for Mer3 and Mlh1 complex formation in yeast meiotic cells, where Mlh2 is necessary (*Figure 3d*).

Importantly, yeast Mer3 and Mlh2, as well as their mouse orthologs HFM1 and PMS1, interacted in a yeast two hybrid assay, showing an evolutionary conservation of the interaction, despite the limited sequence homology between yeast and mammalian Mlh2 (*Campbell et al., 2014*) (*Figure 4—figure supplement 3a,b*).

## The IG-like domain of Mer3 is critical for the Mlh2 interaction

We next sought to obtain a mutant of Mer3 that would specifically alter the Mer3-MutLβ interaction. Since Mlh2 seems important for this interaction in vivo, we focused on a mutant of Mer3 that would specifically alter the direct Mer3-Mlh2 interaction. The Mer3 IG-like domain appeared important for Mer3-Mlh2 interaction because: (1) Mer3 protein lacking IG-like domain no longer interacted with Mlh1 in vivo (*Figure 3c* and *Figure 3—figure supplement 1c*), (2) the predicted IG-like domain of mouse HFM1 was sufficient for the interaction with PMS1 in the yeast two-hybrid assay (*Figure 4—figure supplement 3b*) and (3) IG-like domains are often involved in protein-protein interactions.

Following principles derived from a statistical analysis of protein complex interfaces (*Andreani et al., 2012*), a search for conserved aminoacids potentially involved in protein-protein interactions on the surface of the Mer3 IG-like domain suggested a site centered on Arg893 (*Figure 4c*). We mutated this positively charged residue into a glutamic acid. Remarkably, the interaction between purified Mlh2 and the mutated Mer3R893E protein was strongly diminished (*Figure 4d*). Importantly, the Mer3R893E mutated protein kept its DNA binding and helicase activity

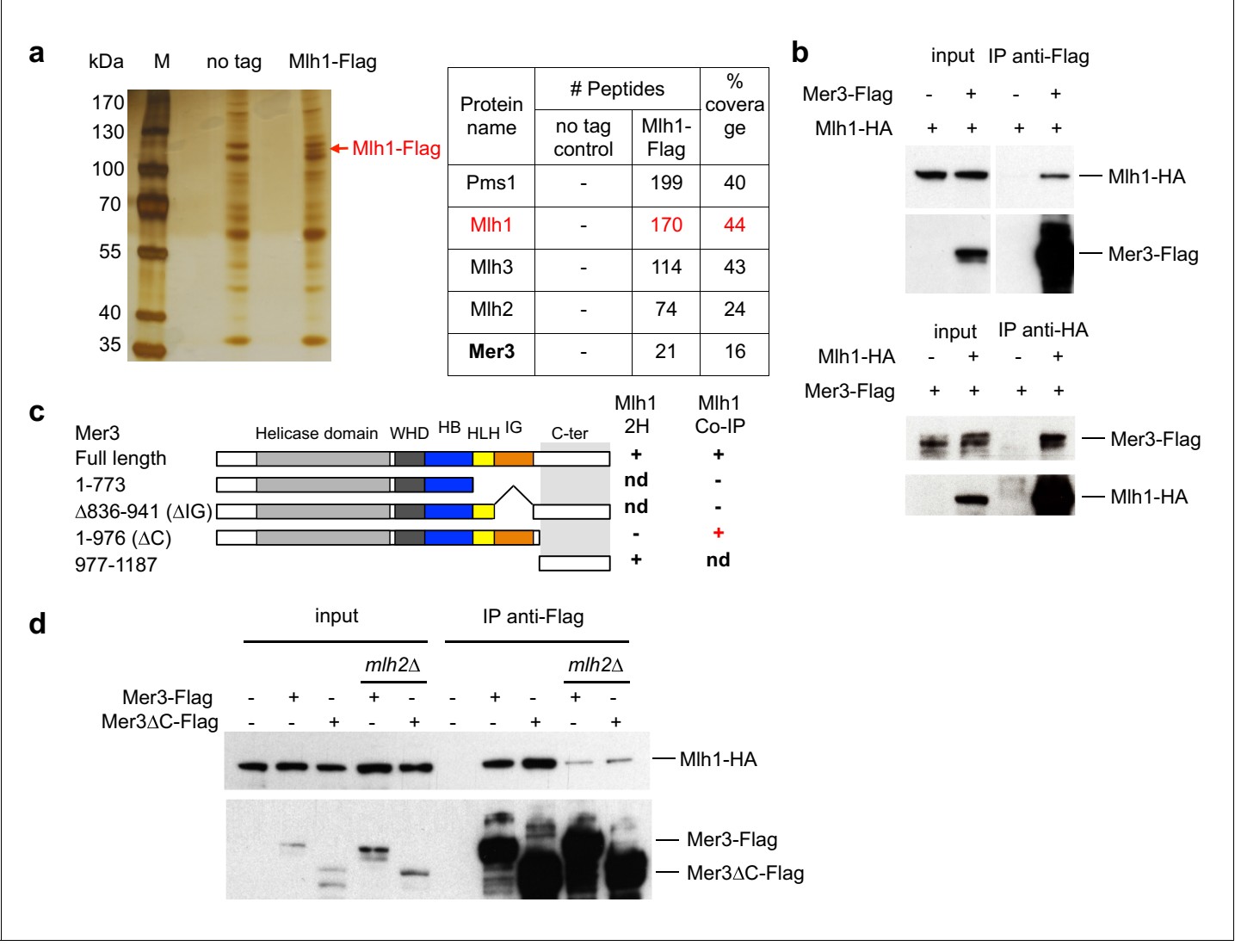

**Figure 3.** Mlh1 interacts with the meiosis-specific Mer3 helicase. (**a**) Affinity pull-down of Mlh1-Flag from cells at 4 hr in meiosis. M: marker, No tag: VBD1311, Mlh1-Flag: VBD1337. Left: silver-stained gel of pulled-down proteins. Right: mass-spectrometry analysis of the most significant proteins pulled-down with Mlh1. One representative experiment is shown. (**b**) Reciprocal co-immunoprecipitation between Mer3-Flag and Mlh1-HA from meiotic cells at 4 hr in meiosis, analyzed by Western blot. Mer3-Flag: VBD1420, Mlh1-HA: VBD1456, Mer3-Flag Mlh1-HA: VBD1454. (**c**) Comparative analysis of two-hybrid interactions and co-IP in meiotic cells. The domain limits are based on Mer3 modeling with the Mer3 family-related Brr2 helicase structure (*Santos et al., 2012*). nd: not determined. (**d**) Co-IP between Mer3-Flag and Mlh1-HA at 4 hr in meiosis, in the presence or absence of MutL$\beta$ component Mlh2. Mlh1-HA: VBD1456, Mer3-Flag Mlh1-HA: VBD1454, Mer3$\Delta$C-Flag Mlh1-HA: VBD1490, Mer3-Flag Mlh1-HA *mlh2$\Delta$*: VBD1550, Mer3$\Delta$C-Flag Mlh1-HA *mlh2$\Delta$*: VBD1552.

The following figure supplement is available for figure 3:

**Figure supplement 1.** Interaction between yeast or mouse Mer3 and Mlh1 and pulldown of Mlh1 and Mlh2 by Mer3.

intact (*Figure 4e* and *Figure 4—figure supplement 4a*). Consistent with the loss of interaction between purified Mer3 and Mlh2 proteins, the Mer3R893E protein was unable to interact with Mlh2 in yeast two hybrid test, while keeping its interaction with Mlh1 (*Figure 4—figure supplement 4b* and *Figure 4—figure supplement 2*).

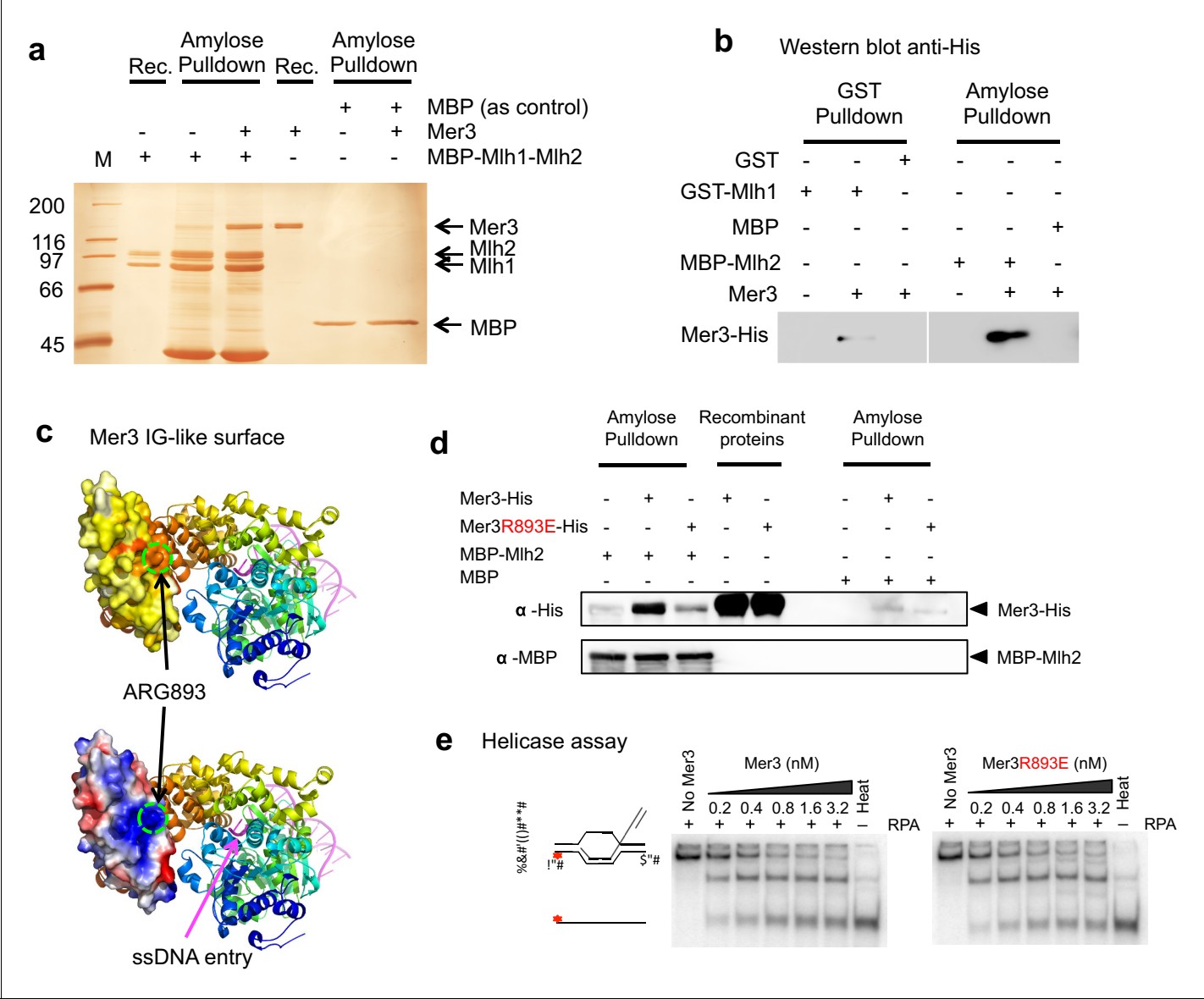

**Figure 4.** Mer3 interacts directly with the Mlh1-Mlh2 (MutL$\beta$) heterodimer. (a) Silver-stained gel showing the direct interaction between purified Mer3 and Mlh1-Mlh2 (MutL$\beta$) complex tagged with MBP on Mlh2. MBP-tagged MutL$\beta$ or MBP were bound to amylose resin and incubated with Mer3-His in the presence of 80 mM NaCl. Proteins were eluted with maltose. (b) Western Blot analysis using anti His antibody showing the pulldown of purified Mer3-His by GST-Mlh1 or by MBP-Mlh2 in the presence of 150 mM NaCl. Both panels are from the same exposure time of the same membrane. (c) Structural model of the IG-like domain of Mer3, based on Brr2 helicase structure (*Santos et al., 2012*). Top diagram: surface of the IG-like domain showing aminoacid conservation, from low (white) to high (red). Bottom: Electrostatic potential indicated by a color code, from positive (blue) to negative (red) charge. The position of Arg893 is indicated. (d) Western Blot showing the pulldown of purified Mer3-His or Mer3R893E-His by MBP-Mlh2 or MBP alone as a control. Pulldown was done in the presence of 150 mM NaCl. (e) Helicase assays with Mer3 or Mer3R893E on labeled D-loop substrate. 'Heat', heat-denatured DNA substrate indicates the position of ssDNA. Assays were performed with or without RPA (20 nM) as indicated.

The following figure supplements are available for figure 4:

**Figure supplement 1.** Purification of yeast Mlh1-Mlh2 and interaction with purified Mer3.
**Figure supplement 2.** Two hybrid interaction between Mer3 and Mlh1 is independent of Mlh2.
**Figure supplement 3.** Mer3 and Mlh2 interact in yeast two-hybrid assays.
**Figure supplement 4.** The Mer3R893E mutation disrupts interaction with Mlh2 but not its DNA binding.

# Mlh2 is recruited to meiotic DSB sites through its interaction with Mer3, independently of mismatches or mismatch recognition factors

We next tested the effect of the Mer3R893E mutation on the Mer3 interaction with MutL$\beta$ in meiotic cells. Consistent with the direct interaction between recombinant proteins, Mer3 and Mlh2 were able to reciprocally co-immunoprecipitate each other and Mer3 co-immunoprecipitated both Mlh1

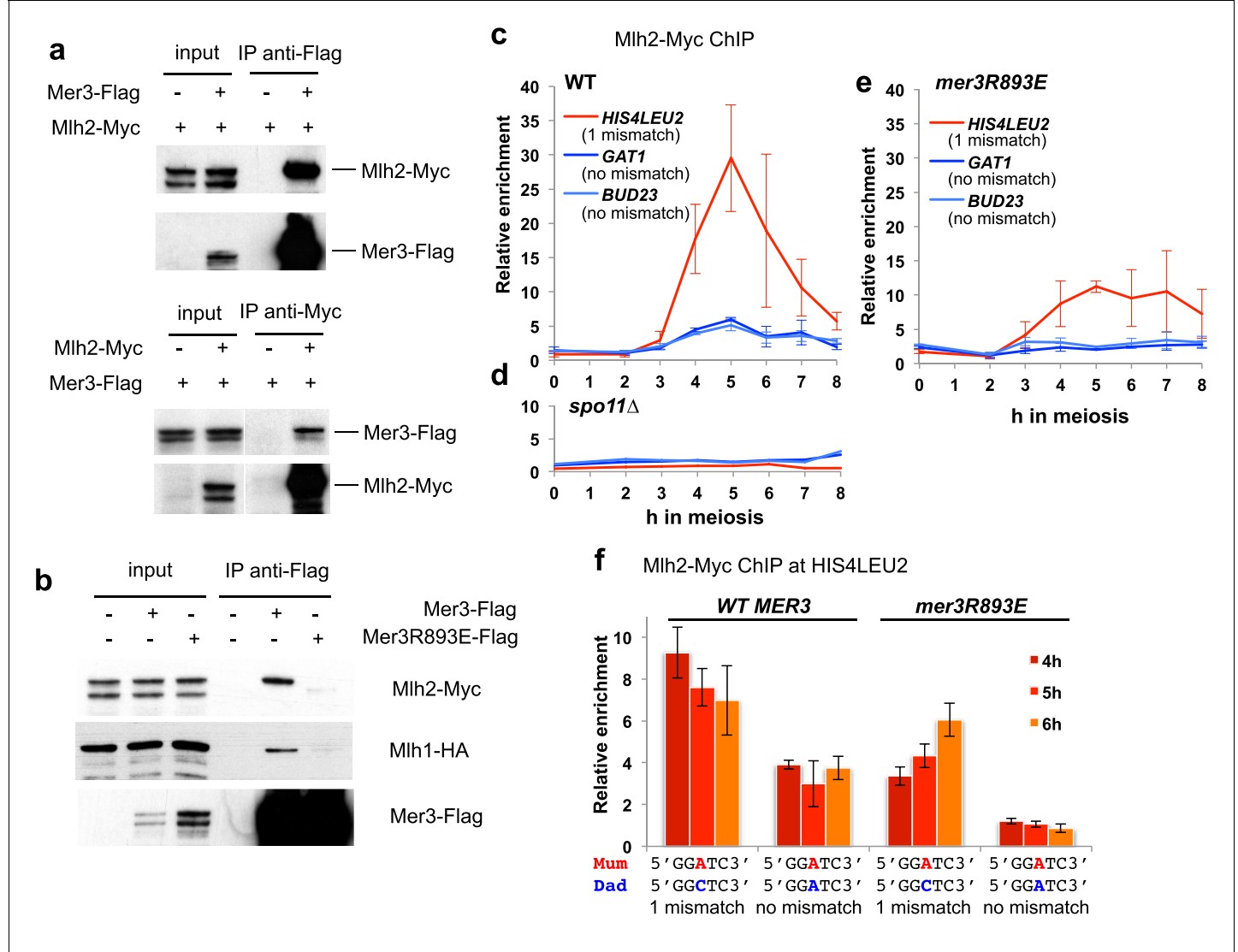

**Figure 5.** Mlh2 binds meiotic recombination intermediates. (**a**) Reciprocal co-IP between Mer3-Flag and Mlh2-Myc from meiotic cells at 4 hr in meiosis, analyzed by Western blot. Mer3-Flag: VBD1420, Mlh2-Myc: VBD1628, Mer3-Flag Mlh2-Myc: VBD1670. (**b**) Co-IP between Mer3- or Mer3R893E-Flag and Mlh2-Myc or Mlh1-HA from meiotic cells at 4 hr in meiosis, analyzed by Western blot. Mlh1-HA Mlh2-Myc: VBD1630; Mer3-Flag Mlh1-HA Mlh2-Myc: VBD1629; Mer3R893E-Flag Mlh1-HA Mlh2-Myc: VBD1681. (**c**) Mlh2-Myc levels at three meiotic recombination hotspots assessed by ChIP and qPCR at the indicated times during a meiotic time-course (VBD1670). (**d**) Same as in (**c**) but in a DSB-deficient *spo11Δ* strain (VBD1702). (**e**) Same as in (**c**) but in the *mer3R893E* strain (VBD1637). (**f**) Effect of polymorphism at the *HIS4LEU2* hotspot on Mlh2 binding. Mlh2-Myc association with the *HIS4LEU2* hotspot was assessed by ChIP at the indicated times in meiosis in strains containing either the wild-type or R893E *mer3* allele, in strains with one base mismatch or without a mismatch at the *HIS4LEU2* hotspot. WT one base mismatch: VBD1670; WT no mismatch: VBD1710; *mer3R893E* one base mismatch: VBD1637; *mer3R893E* no mismatch: VBD1706. (**c**), (**e**) and (**f**): Values are the mean ± SEM from two independent experiments.

The following figure supplement is available for figure 5:

**Figure supplement 1.** Polymorphism at the *HIS4LEU2* hotspot triggers Mlh2 recruitment by Msh2.

and Mlh2 (*Figure 5a and b*). The Mer3R893E protein instead no longer interacted with Mlh2 or Mlh1 in meiotic cells, confirming the critical role of the Mer3-Mlh2 interaction in a complex formation between Mer3 and the MutLβ heterodimer in vivo (*Figure 5b*).

Overall, our data demonstrate that in yeast the interaction between Mer3 and Mlh1 is weak, whereas the interaction between Mer3 and Mlh2 is stronger, and required for the complex formation between Mlh1 and Mer3 in vivo. In contrast, for mouse proteins, the interaction between HFM1 and MLH1 is very strong, compared to that between HFM1 and PMS1. In both systems however, Mer3/HFM1 forms a complex with the MutLβ heterodimer.

As anticipated from a protein interacting with Mer3, Mlh2 associated with meiotic DSB hotspot sites, at the time of recombination, in a Spo11 DSB-dependent manner (*Figure 5c and d*). Remarkably, Mlh2 association with the *BUD23* and *GAT1* hotspots no longer occurred in the *mer3R893E* mutant, implying that Mer3 recruits Mlh2 to these sites (*Figure 5e*). Surprisingly, Mlh2 recruitment to the *HIS4LEU2* hotspot was reduced, but not eliminated (*Figure 5e*). The *HIS4LEU2* hotspot contains one base polymorphism between the two alleles (*Hunter and Kleckner, 2001*; *Martini et al., 2006*), which potentially creates a mismatch during interhomolog recombination and the formation of a D-loop intermediate (*Figure 5f*). Interestingly, eliminating the polymorphism at the *HIS4LEU2* hotspot reduced Mlh2 recruitment, which was abolished when combined with the *mer3R893E* mutation (*Figure 5f*). Consequently in *msh2Δ* cells, Mlh2 recruitment to the two hotspots *BUD23* and *GAT1* was not affected, but it was reduced at the polymorphic *HIS4LEU2* hotspot (*Figure 5—figure supplement 1*).

All together, our results show that the mode of recruitment of Mlh2 through interaction with Mer3 does not require a mismatch. If a mismatch is formed during recombination between heteroalleles, Mlh2 is recruited, independently of Mer3, most likely by Msh2.

## Mlh1-Mlh2 preferentially recognizes D-loops and related branched DNA structures and lacks endonuclease activity

Unlike Mlh3 and Pms1, Mlh2 lacks the conserved endonuclease motif characteristic for the eukaryotic MutL family of proteins (*Campbell et al., 2014*; *Kadyrov et al., 2007*; *Ranjha et al., 2014*; *Rogacheva et al., 2014*). In agreement, Mlh1-Mlh2 was deficient in an endonuclease assay (*Figure 6a*). We next investigated whether the MutLβ complex binds various DNA structures. Mlh1-Mlh2 preferred binding to pre-Holliday junction (HJ) structures, namely D-loop and bubbled substrate, whereas the binding affinity to nicked HJ, Y-structure and HJ was lower (*Figure 6b* and *Figure 6—figure supplement 1*). Binding to dsDNA was ~8.5 times lower than to D-loops, while ssDNA was almost not bound. We next investigated the DNA binding specificity of Mer3. Previous studies indicated that Mer3 binds equally double-stranded DNA and Holliday Junctions (*Nakagawa and Kolodner, 2002a*). We determined that purified Mer3 showed a striking preference for binding D-loop structures over all other substrates, and efficiently unwinds D-loops, consistent with its proposed role on this intermediate (*Figure 6c*). The stronger affinity of Mer3 to D-loops compared to that of MutLβ is also consistent with our findings that Mer3 is required for the recruitment of MutLβ to these intermediates in vivo. Collectively, these results suggest a non-catalytical function of MutLβ on early recombination intermediates.

## Mlh2 meiotic function is dependent on its interaction with Mer3

We next investigated the role of the Mer3-Mlh2 interaction in meiotic recombination. *mlh2Δ* or *mer3R893E* cells progressed through the first meiotic division with a delay compared to a wild-type strain. In addition, this delay was dependent of the presence of DSB (*Figure 7a*). In the SK1 background, both mutants formed viable spores and showed normal CO frequency at the *HIS4LEU2* hotspot (*Figure 7b* and *Figure 7—figure supplement 1*). However, in a hybrid SK1/S288C background, spore viability was significantly reduced in both *mlh2Δ* and in *mer3R893E* mutants (*Figure 7b* and *Figure 8—figure supplement 1d*). Together, the DSB-dependent delay of meiotic progression and the loss of spore viability indicate that *mlh2Δ* and *mer3R893E* mutants have a meiotic defect due to a defect in DSB repair.

The ZMM proteins Zip4 and Msh4 are important for CO formation, and their deletion leads to a decrease in both CO frequencies and spore viability. Strikingly, both the *mlh2Δ* and the *mer3R893E* mutations improved the spore viability of the *zip4Δ* and *msh4Δ* mutants (*Figure 7c*), which had

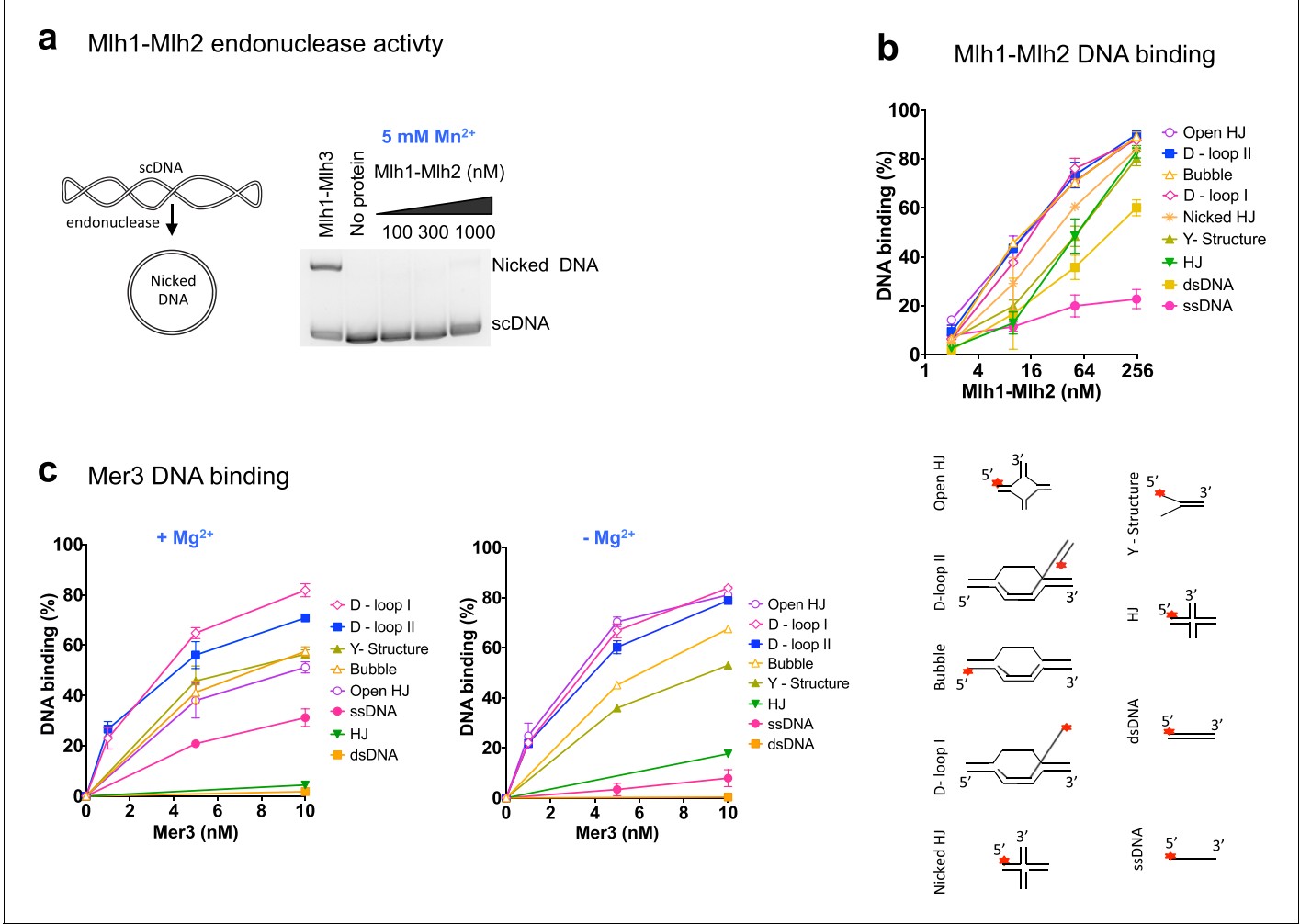

**Figure 6.** Recombinant Mer3 and MutL$\beta$ preferentially bind DNA substrates mimicking early recombination intermediates. (**a**) Nuclease assay was performed with Mlh1-Mlh2 (as indicated) or Mlh1-Mlh3 (300 nM) on the indicated super-coiled circular plasmid DNA substrate (scDNA). Mlh1-Mlh3 is shown as a positive control producing nicked circular DNA (**b**) Quantitation of electrophoretic mobility shift assays with Mlh1-Mlh2 and various oligonucleotide-based DNA substrates in the presence of magnesium (2 mM). Values are the mean ± SEM from two independent experiments. (**c**) Electrophoretic mobility shift assays with Mer3 and the indicated DNA substrates, in the presence of either 2 mM magnesium (+Mg$^{2+}$) or 3 mM EDTA (−Mg$^{2+}$). Values are the mean ± SEM from two to tree independent experiments.

The following figure supplement is available for figure 6:

**Figure supplement 1.** DNA binding specificities of Mlh1-Mlh2.

already been noticed for the *mlh2Δ msh4Δ* mutant compared to *msh4Δ* (*Abdullah et al., 2004*). Increased spore viability may result from an increased capacity of these mutants to make crossovers. However, the deletion of *MLH2* did not lead to a detectable increase of CO frequency at the *HIS4LEU2* hotspot in *zip4Δ* or *msh4Δ* mutants (*Figure 7—figure supplement 1*). This indicates that either the spore viability increase is not related to an increase in CO, or that a subtle increase below the detection limit of our assay may be sufficient to increase spore viability. Noteworthy, our results show that the meiotic phenotypes of *mlh2Δ* and *mer3R893E* mutants are identical, indicating that the meiotic functions of Mlh2 are exerted through its interaction with Mer3.

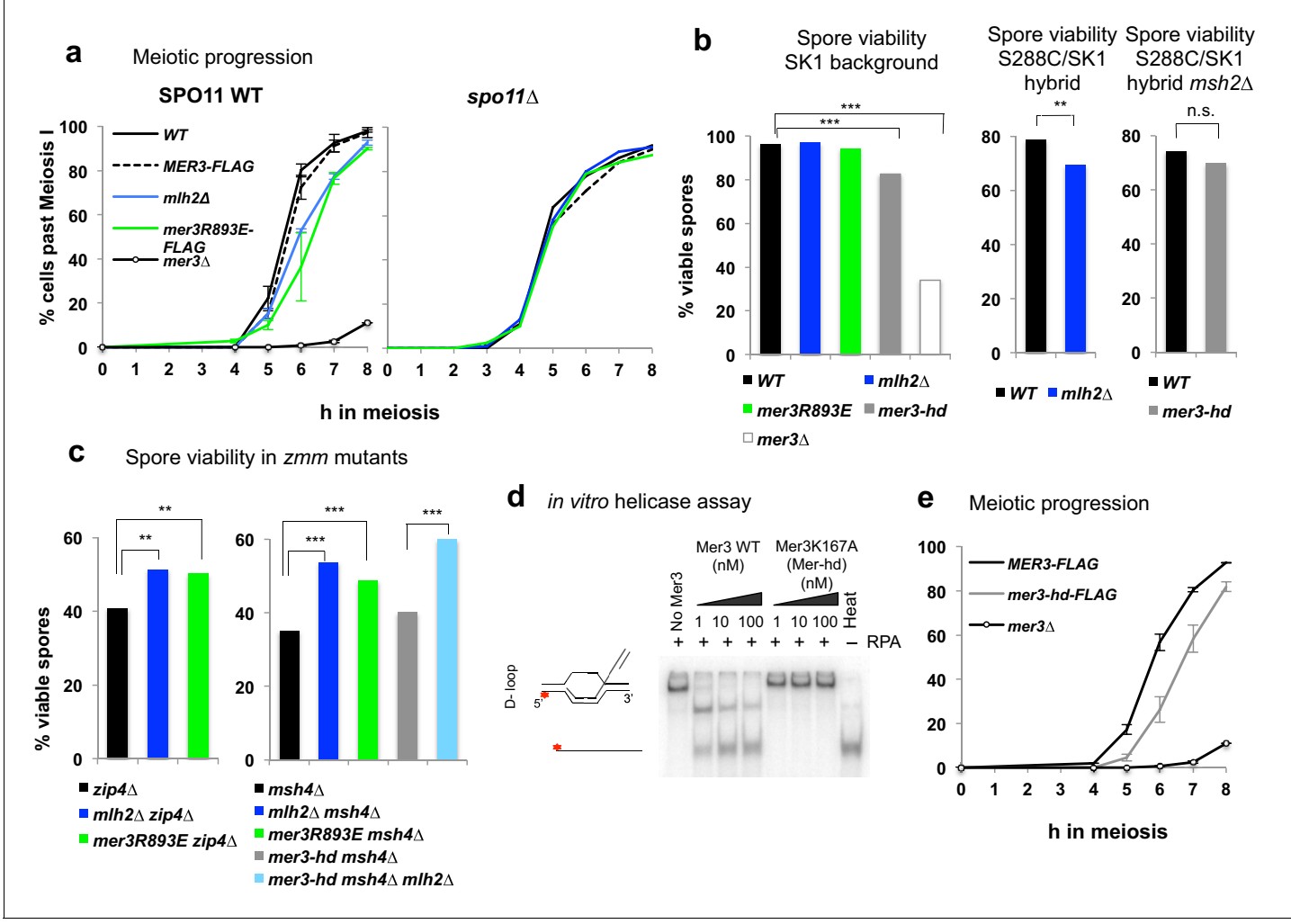

**Figure 7.** Disruption of the Mlh2/Mer3 interaction improves spore viability of *zmm* mutants. (a) Meiotic progression as assessed by DAPI staining of strains with the indicated genotype: WT (VBD1311), Mer3-Flag (VBD1420), *mlh2Δ* (VBD1631), Mer3R893E-Flag (VBD1635), *mer3Δ* (VBD1414), *spo11Δ* (VBD1382), *spo11Δ* Mer3-Flag (VBD1794), *spo11Δ mlh2Δ* (VBD1796), *spo11Δ* Mer3R893E-Flag (VBD1795). (b,c) Spore viability assays of strains with the indicated genotype. Fisher's exact test, **p<5.10$^{-3}$; ***p<5.10$^{-5}$; n.s.: p>0.05. Numbers of tetrads dissected and strains names are in *Supplementary file 1*. (d) Helicase assays with recombinant Mer3 or Mer3K167A in the presence of 20 nM RPA on a D-loop substrate. 'Heat', heat-denatured DNA substrate indicates the position of ssDNA. (e) Meiotic progression as assessed by DAPI staining of strains with the indicated genotype: Mer3-Flag (VBD1420), Mer3-hd-Flag (VBD1750) or *mer3Δ* (VBD1414). (a), left panel and (e): Values are the mean ± SEM from two independent experiments.

The following figure supplement is available for figure 7:

**Figure supplement 1.** Effect of the Mer3-Mlh2 interaction on CO frequency at the *HIS4LEU2* hotspot.

## Mlh2 limits the extent of sequences involved in meiotic recombination intermediates through its interaction with Mer3

Since abolishing the interaction between Mer3 and Mlh2 compensates partially the viability defects of *zmm* mutants, we asked if and how it affects recombination intermediates. To get a precise genome-wide view of recombination intermediates, we sequenced all the eight DNA strands resulting from meioses of S288C*SK1 hybrid diploids. The *msh2Δ* mutation present in this hybrid allows the detection of virtually all interhomolog recombination events, since most DNA heteroduplexes (hDNA) formed during recombination between polymorphic parental sequences are maintained due to MMR deficiency, whereas they would lead to gene conversions or restorations of the parental

alleles in the *MSH2* background (*Figure 8a*) (*Martini et al., 2011*). For simplicity, we will refer to all the strand transfer events that we observed, either hDNA or gene conversions, as 'gene conversion tracts'. In this hybrid diploid, CO frequency appeared reduced by about 25% in *mlh2Δ* and in *mer3R893E*, although these measurements have to be taken with caution since they are issued from 2 (WT and *mlh2Δ*) or 3 (*mer3R893E*) meioses of each genotype (*Figure 8—figure supplement 1a*). COs promoted by the ZMM pathway exhibit interference, i.e. the tendency to be evenly spaced,

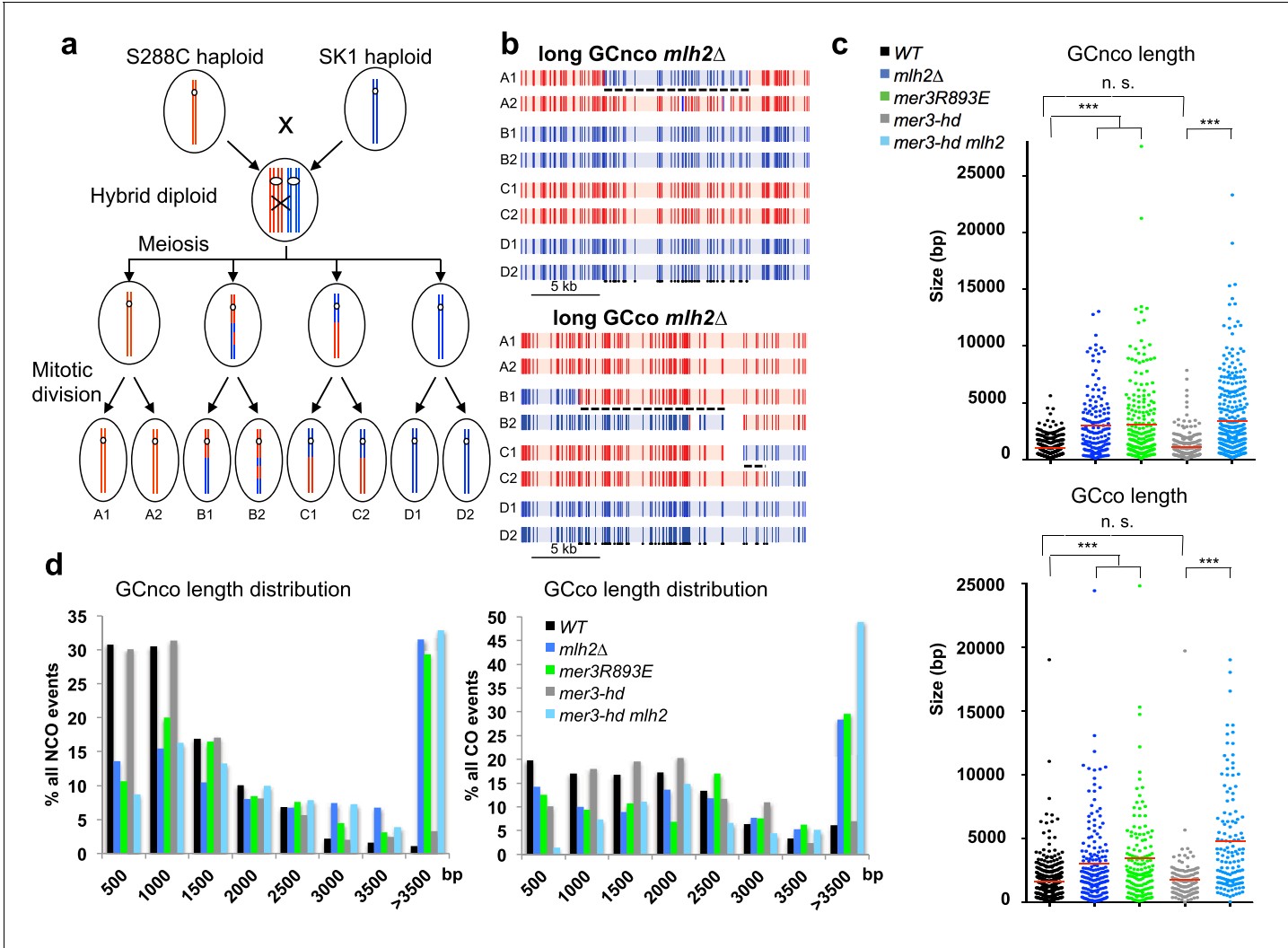

**Figure 8.** The Mlh2/Mer3 interaction limits the extent of gene conversions genome-wide. (a) Scheme of the experimental system to measure genome-wide recombination events. After meiosis, the four haploid spores were allowed to perform one mitosis and micromanipulated, in order to sequence DNA of the two daughters, allowing the recovery of the 8 DNA recombined strands from the initial diploid cell. (b) Blow up of NCO and CO events of a *mlh2Δ* meiosis, showing very long hDNA tracts (referred to as GC, or gene conversions) more frequent in this mutant. The 8 strands are shown. Each vertical bar represents a nucleotide polymorphism between the two strains (blue: of SK1 origin; red: of S288C origin). (c) Gene conversion lengths among NCO and CO events of meioses from the indicated genotype. The horizontal bar indicates the mean value of all events from the meioses (4 meioses of WT, and 2 meioses of each other relevant genotype). Wilcoxon rank sum test, ***p<5.10$^{-5}$. n.s.: p>0.05 (d) Distribution of gene conversion lengths among NCO and CO events of meioses each from the indicated genotype.

The following figure supplements are available for figure 8:

**Figure supplement 1.** CO and NCO numbers and other events in the hybrid strains.

**Figure supplement 2.** Formation of symmetrical heteroduplex upon D-loop extension in the direction opposed to DNA synthesis.

whereas COs promoted in the absence of the ZMM pathway do not (*Hunter, 2015*). We estimated CO interference from the distribution of the inter-CO distances measured in our genome-wide experiments, using a gamma distribution model as previously described (*Chen et al., 2008*). Consistent with previous findings, we obtained a gamma value of 2.3 for wild-type. *mlh2Δ* and *mer3R893E* cells showed gamma values of 2.5 and 1.9, respectively, indicating that CO interference still occurs, as opposed to values between 1.1 and 1.2 observed in true *zmm* mutants (*Chen et al., 2008*). This indicates that *mlh2Δ* and *mer3R893E* mutants still use the ZMM pathway for CO formation as wild-type cells (*Figure 8—figure supplement 1b*).

Strikingly, the length of both CO- and NCO- associated conversion tracts strongly increased in *mlh2Δ* and *mer3R893E* mutants (*Figure 8b*). Tracts associated with a CO went from a median length of 1.0 kb in wild-type to 2.1 kb (*mlh2Δ*) and 2.3 kb (*mer3R893E*) and tracts associated with NCO went from 0.8 kb to 2.3 kb (*mlh2Δ*) and 1.7 kb (*mer3R893E*) (*Figure 8c*). In particular, very long tracts larger than 10 kb occurred in both mutants and a large proportion of events spanned longer than 3.5 kb in length (*Figure 8d*).

Apart from the extended length of gene conversion tracts, no striking change in the nature of the events occurred in the *mlh2Δ* and *mer3R893E* mutants. The two major categories of NCO events, defined depending on the associated strand transfer patterns, occurred with the same frequency as in wild-type and showed a similarly increased length in both mutants (*Figure 8—figure supplement 1c*) (*Martini et al., 2011*). We conclude that the interaction between Mer3 and Mlh2 is essential to limit the length of conversion tracts genome-wide, both at CO- and NCO-designated events, without affecting the repair outcome.

Genetic and biochemical evidence suggests that Mer3 extends early recombination intermediates to stabilize them to promote CO formation (*Börner et al., 2004*; *Mazina et al., 2004*). We wondered if the long extension of conversion tracts seen in the absence of Mer3-Mlh2 interaction is due to the helicase capacity of Mer3 itself. Based on its in vitro helicase activity, Mer3 should extend the D-loop in the direction opposed to DNA synthesis, resulting in a type of recombination product called symmetrical heteroduplex (*Figure 8—figure supplement 2*) (*Mazina et al., 2004*). However, neither the frequency nor the length of these events, very rare in wild-type, was increased in the mutants [WT: 560 bp (11 events in 4 meioses), *mlh2Δ*: 644 bp (7 events in 2 meioses), *mer3R893E*: 434 bp (8 events in 2 meioses)], arguing against Mer3 extending D-loops in this direction in the absence of interaction with Mlh2.

We therefore directly tested the involvement of the Mer3 helicase activity in the increase of conversion tract length by making a strain containing the *mer3K167A* (*mer3-hd*) mutation, which abolishes Mer3 helicase activity (*Figure 7d*) (*Nakagawa and Kolodner, 2002b*). Surprisingly, genome-wide mapping of recombination events clearly showed that (1) the length of conversion tracts was identical to wild-type in the *mer3-hd* mutant, (2) the length of conversion tracts was still strongly increased when deleting *MLH2* in the *mer3-hd* mutant, showing that the helicase of Mer3 is not involved in the increased length of events seen in the absence of Mer3-Mlh2 interaction (*Figure 8c and d*). In further support of this, the *mer3-hd* mutation did not impact on the increased spore viability of the *mlh2Δ msh4Δ* double mutant, unlike the *mer3Δ* mutation (*Figure 7c* and *Supplementary file 1*). We conclude that the phenotypes of *mlh2Δ* (increased viability of *zmm* mutants and increased length of gene conversion tracts) are not due to Mer3 helicase activity. Surprisingly, the *mer3-hd* mutant also still displayed an almost intact 'ZMM' function, with high spore viability and only a short delay in meiotic progression, as opposed to the *mer3Δ* mutant (*Figure 7b and e*) (*Jessop et al., 2006*). CO numbers in the *mer3-hd* hybrid mutants were decreased compared to the wild-type or the single *mlh2Δ* mutant, but less than would be expected from a *zmm* mutant as determined from previous studies (*Chen et al., 2008*; *Oke et al., 2014*) (*Figure 8—figure supplement 1a*). Mer3 thus fulfills its early function (controlling gene conversions length) and most of its later ZMM functions (meiotic progression, spore viability and CO formation) independently of its helicase activity.

## Discussion

In this study, we show that the MutLβ complex is specifically recruited to meiotic recombination sites by the Mer3 helicase, and shows the unique ability to limit the length of gene conversions genome-wide. This regulation may be conserved in mammals since the Mer3-MutLβ interaction is conserved

between the mouse proteins. This has important implications for the control of genome diversity created by gene conversion in meiosis.

## Model

The recombination signature of the *mlh2Δ* or *mer3R893E* mutants shows that hDNA is extended mainly in the direction of DNA synthesis. We show that the MutL*β*-Mer3 complex stops D-loop extension and associated DNA synthesis, limiting gene conversion tract length. Strikingly, even if the Mer3 helicase activity is not involved, the long hDNA extension seen in *mlh2Δ* seems to require a structural function of Mer3, since *mlh2Δ* does not improve spore viability of the *zip4Δ* mutant in the absence of Mer3 (**Supplementary file 1**).

Based on our findings, we propose the following model (**Figure 9**): (1) Mer3 binds D-loop intermediates, thanks to its preferential affinity for this substrate and induces a structural change making it able to migrate and (2) Mer3 recruits MutL*β*, which also preferentially binds D-loop structures. The formed Mer3-MutL*β* complex acts as a lock to physically block the overextension of recombination

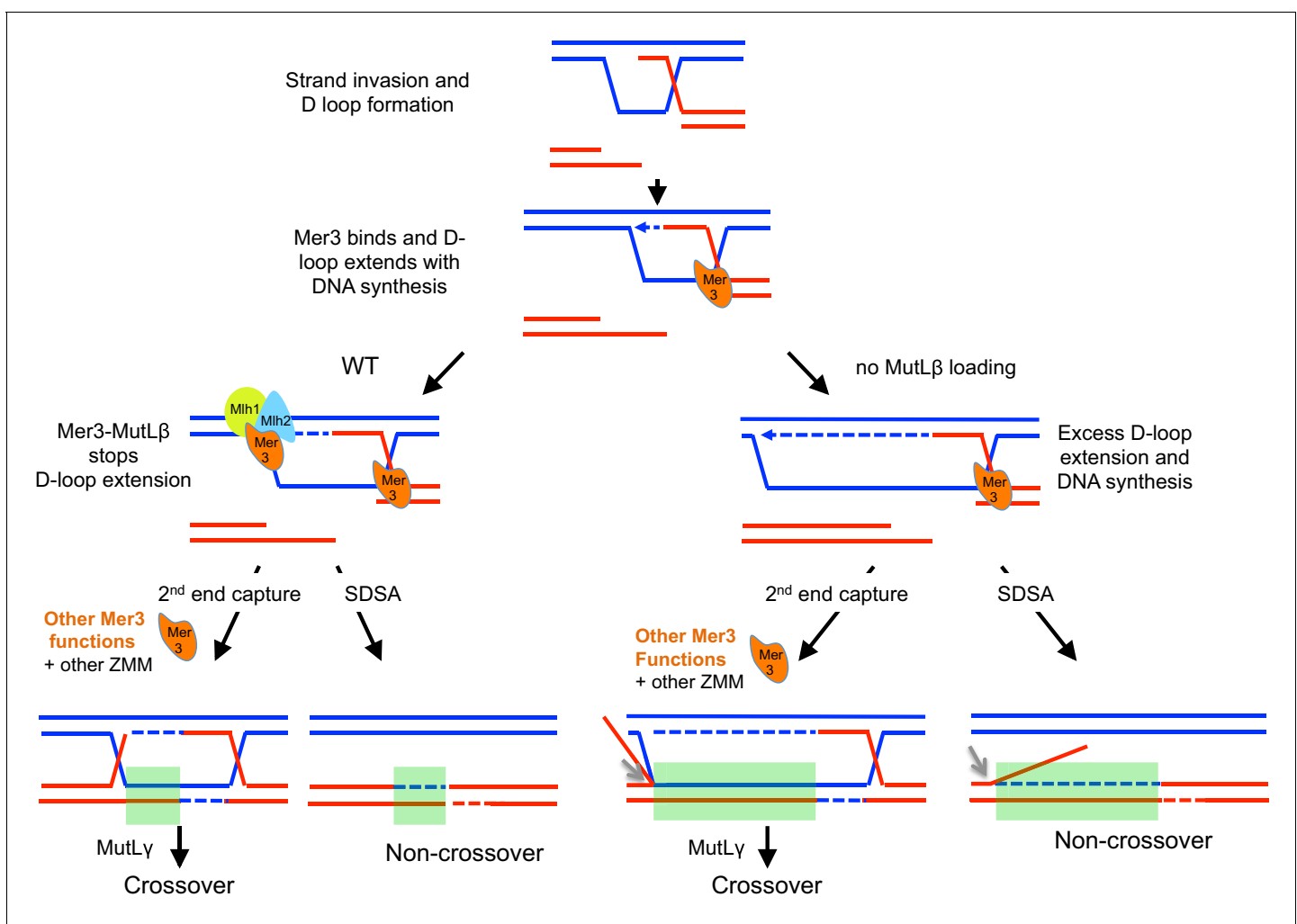

**Figure 9.** Model for the actions of Mer3 and MutL*β* on recombination sites. Following DSB formation and strand invasion, Mer3 (in orange) binds the resulting D-loop and DNA synthesis begins (dotted blue arrow). MutL*β* (light blue and yellow complex) interaction with Mer3 then acts as a lock to stop D-loop extension in DNA synthesis direction (left panel). In the absence of Mlh2, D-loop will extend significantly further in the DNA synthesis direction (right panel). After this step, Mer3 is expected to have other, procrossover functions, partly dependent on its helicase activity, acting together with the other ZMM proteins and MutL*γ*. Grey arrows indicate that endonuclease activity should remove the flap generated by overextension of the D-loop in the absence of MutL*β*. The light green rectangles indicate the gene conversion tracts, longer in the absence of MutL*β*.

intermediates. (3) Mer3 then exerts its later functions, involving its helicase activity in conjunctions with the other ZMM proteins to form crossovers. In the absence of Mlh1-Mlh2, the Mer3-bound D-loop is over-extended by the action of DNA polymerase. The next steps proceed as in wild-type involving later functions of Mer3 for CO formation. The resulting meiotic products show a large increase of gene conversions tracts length at all recombination sites.

## Mer3 has several distinct roles in meiotic recombination

Null mutants of *MER3* show meiotic defects typical for the ZMM proteins. Specifically COs, but not NCOs, are reduced, likely as a result of less stable early recombination intermediates (*Börner et al., 2004*; *Jessop et al., 2006*). These results placed Mer3 as a procrossover factor acting in the main CO pathway that displays CO interference. However, since both CO- and NCO-associated conversion tracts are longer in the *mer3R893E* mutant, and since Mer3 preferentially binds D-loop intermediates, a common precursor to all recombination events, our results uncover a new early function of Mer3. This infers that Mer3 has a function on D-loop intermediates, irrespective of their fate. This Mer3 function is independent of its 'ZMM' and procrossover functions, since the *mer3R893E* mutation does not affect meiotic progression or CO frequencies. Previous cytological data in *Sordaria macrospora* and in rice showed the presence of Mer3 foci at early leptotene and in much higher numbers than COs, also suggesting an early function of Mer3 in these organisms (*Storlazzi et al., 2010*; *Wang et al., 2009*).

Our results showing that Mer3 fulfills most of its early and late functions independently of its helicase activity appear to contrast with previous reports that proposed that the helicase of Mer3 is important to extend and stabilize recombination intermediates at future CO sites (*Mazina et al., 2004*; *Nakagawa and Kolodner, 2002b*). Although the Mer3 helicase activity seems to be involved in promoting normal CO frequency, we reveal that Mer3's ZMM function is largely non-catalytical, and may involve the recruitment of additional ZMM proteins to stabilize recombination intermediates (*Börner et al., 2004*).

## A molecular function for the MutLβ complex, independent of mismatch recognition

MutL*β* is recruited to recombination sites through mismatch recognition and Mer3. However, these recruitment modes are not redundant. Recruitment through mismatches does not restrain GC tract length as seen in the *mer3R893E* mutant (*Figure 5f*), and MutL*β* efficiently restrains GC tract length in the absence of Msh2. This shows that MutL*β* participates in two distinct processes at recombination sites. The novel structural rather than catalytical role of MutL*β* that we described here is in contrast with its MutL*α* and *γ* relatives, which require nuclease activities for their functions (*Kadyrov et al., 2007*; *Nishant et al., 2008*). In addition, our findings provide a role for MutL*β*, which has been elusive for almost 30 years, and show that MutL*β* is the only MutL complex that functions independently of a MutS-related heterodimer (*Manhart and Alani, 2016*). Interestingly, a recent comparative genomics study revealed that Mlh2 is lost concomitantly with the ZMM proteins in the *Lachancea* budding yeast clade, consistent with the functional connection we describe here (*Vakirlis et al., 2016*). It is also interesting that in vegetative budding yeast cells, *mlh2Δ* mutants show higher resistance to DNA damaging drugs, which led to the proposal that MutL*β* could inhibit Rad51-dependent recombination (*Durant et al., 1999*). Perhaps a mechanism similar to the one we describe for meiosis is at play, in which MutL*β* limits the formation of early recombination intermediates. Its deletion would favor the stabilization of early recombination intermediates and homologous recombination. It is worth noticing that in human cells, PMS1 (the Mlh2 homolog) interacts with several helicases (*Cannavo et al., 2007*). The functionality of these interactions has not been determined yet.

Finally, our study reinforces the tight links that exist between homologous recombination and mismatch repair related MutL complexes. It is fascinating that two of the three MutL complexes are an integral part of the meiotic homologous recombination process. Since MutL*β* and MutL*γ* both contain Mlh1, our results also reveal an unprecedented early role of Mlh1 at meiotic recombination intermediates, within the MutL*β* complex. Mlh1 is classically used as a cytological marker of crossovers in several organisms such as mammals, where it forms foci on synapsed homologs, but given

the conserved interaction between Mer3 and MutL$\beta$ in mammals, careful examination may reveal earlier Mlh1 foci on meiotic chromosomes, as early as in leptotene/zygotene stage.

## Control of the length of gene conversions: a new regulatory step of meiotic recombination

We did not detect a pronounced effect of the over-extended recombination intermediates on CO frequency. Regarding the proposed role of extending the D-loop into a single-end intermediate for promoting COs, one could have expected *mlh2Δ* or *mer3R893E* mutant to produce more COs, because a longer intermediate would be resistant to disruption by helicases (*De Muyt et al., 2012*). However, CO number in budding yeast and other organisms is highly regulated, especially by CO interference, which limits crossovers (*Hunter, 2015*). Since interference is still fully active in the *mlh2Δ* and *mer3R893E* mutants, we infer that the CO decision is made before or irrespective of the extension of joint molecules in the absence of Mlh2.

There are other examples of mutants, which affect the length of D-loops but have no effect on CO frequency. An *exo1* nuclease-dead mutant displayed reduced DNA end resection length, which was expected to form shorter, less stable D-loops, but in contrast no change in CO frequency was observed (*Zakharyevich et al., 2010*). This 'buffering' effect may be related to the phenomenon of crossover homeostasis, which maintains CO numbers, even in compromised recombination contexts (*Lao et al., 2013*; *Martini et al., 2006*).

Why would meiotic cells limit the extent of sequences involved in gene conversions? Strikingly, gene conversion tracts are much longer in *S. cerevisiae* somatic recombination than in meiosis (*Yim et al., 2014*), consistent with the meiosis-specific nature of the negative regulation by Mer3-MutL$\beta$. Compared to somatic recombination, a specificity of meiosis is the simultaneous induction of high numbers of programmed DSBs, potentially very dangerous for genome integrity and that need to be controlled. It may be detrimental for pairing and/or chromosome disentangling to expose too long DNA sequences, or to run into functional elements such as centromeres or the sequences that attach chromatin loops to the meiotic chromosome axis. In *Sordaria* meiosis, Mlh1 mutants show an increased frequency of entanglements, suggesting a function at an earlier stage than crossover resolution (*Storlazzi et al., 2010*). These may be related to the over-extension of D-loops that we see in the absence of MutL$\beta$ and its interaction with Mer3.

Interestingly, limiting gene conversion specifically in meiosis may also serve the purpose of avoiding, in the long term, breaking of ancestral linkage groups of favorable combination of alleles and homogenizing alleles in a population that may occur when DSB formation is more frequent on one allele (*Figure 1*; reviewed in (*Cole et al., 2012*). In addition, in human and mice, recombination is initiated at site-specific sequences, which could be frequently erased by conversion tracts, leading to the rapid extinction of hotspots, in the absence of a limiting mechanism (*Baudat et al., 2013*; *Cole et al., 2014*).

## Materials and methods

### Yeast strains and media

All yeast strains are derivative of the SK1 background except otherwise stated and are listed in *Supplementary file 2*. A table that links strains to figure panels is provided in *Supplementary file 3*.

For synchronous meiosis, cells were grown in SPS presporulation medium and transferred to 1% potassium acetate with vigorous shaking at 30°C as described (*Murakami et al., 2009*). For S288C*SK1 hybrid sporulation, cells were grown overnight in YPA presporulation medium, and then transferred to liquid 1% potassium acetate. Tetrads were dissected, and daughter and mother cells from each spore were separated by micromanipulation as described (*Martini et al., 2011*). For all strains, spore viability was measured after sporulation on solid sporulation medium for two days at 30°C.

### Yeast strains construction

Yeast strains were obtained by direct transformation or crossing to obtain the desired genotype. Site directed mutagenesis, internal and C-terminal deletions were introduced by PCR. All

transformants were confirmed using PCR discriminating between correct and incorrect integrations and sequencing for epitope tag insertion or mutagenesis.

For Mlh1 internal tagging, the indicated HA6 or His6-Flag3 tag sequence, flanked on each side by the GGGGSGGGGS linker sequence, was inserted between aminoacids 710 and 713 of Mlh1. Mlh2 was C-terminally tagged with 13 copies of Myc (*Longtine et al., 1998*). The tagged Mlh2-Myc protein was functional since a *zip4Δ* Mlh2-Myc strain showed the same spore viability as a *zip4Δ* mutant, in contrast to a *zip4Δ mlh2Δ* mutant (*Supplementary file 1*). The His6-Flag3 epitope, preceded by a GGGGSGGGGS linker sequence, was fused to the C-terminus of *MER3*, using plasmid pU6H3FLAG (a gift from Kunihiro Otha).

The *mer3R893E* mutation in the S288C*SK1 hybrid was obtained by CRISPR-Cas9 mediated cleavage, using a plasmid encoding Cas9 and expressing a guide RNA targeted to the *MER3* gene (plasmid generously provided by G. Zhao and B. Fetcher) transformed together with a healing *mer3* fragment containing the R893E mutation.

## Mutation analysis

Mutation rates in the presence of tagged *MLH1* alleles were estimated by measuring the spontaneous reversion rate at the *lys2::InsE-A14* locus in strains derived from E134 (*Shcherbakova and Kunkel, 1999*; *Tran et al., 1997*). Three single colonies from three independent transformants of the same genotype were grown to stationary phase in liquid YPAD medium and plated onto YPAD or selective medium lacking lysine for revertant count.

## Two-hybrid analyses

*MLH1*, *MLH2* and *MLH3* ORFs were PCR-amplified from SK1 genomic DNA. *MER3* cDNA sequence was amplified from SK1 genomic DNA by fusion of 2 PCR products eliminating *MER3*'s intron. Full length mouse HFM1, MLH1, MLH3 cDNA were amplified from plasmids (obtained from Genscript, Genocopia and Genocopia, respectively) and PMS1 was PCR-amplified from mouse testis cDNA, a gift from J. Barau and D. Bourc'his. PCR products were cloned in pDNOR Gateway plasmids and subcloned in Gateway plasmids derived from the two hybrid vectors pGADT7 (GAL4-activating domain) and pGBKT7 (GAL4-binding domain) creating N terminal fusions (gift from M. Grelon). Point mutations, truncations or internal deletions were introduced by PCR. All plasmid inserts were sequenced.

Yeast strains and 2 hybrid experiments were performed as in (*Kumar et al., 2010*). Interaction is defined compared to the growth seen in the negative control consisting of the combination between the GAL4BD-bait protein in the presence of GAL4AD-only ('empty' on the figures). Any combination that grows better than this control on the selective media is considered as an interaction

## Flag-affinity pull-down and mass spectrometry analysis

$2.10^{10}$ cells were harvested, washed two times with ice-cold TNG buffer (50 mM Tris/HCl pH 8; 150 mM NaCl, 10% Glycerol; 1 mM PMSF; 1X Complete Mini EDTA-Free (Roche)) and flash-frozen in liquid nitrogen. Frozen cells were mechanically ground in liquid nitrogen. The resulting powder was resuspended in 50 mL of lysis buffer (50 mM Tris/HCl pH 7.5; 1 mM EDTA; 0.5% NP40; 10% glycerol; 300 mM NaCl). The lysate was cleared by centrifugation at 8000 g for 10 min and then incubated with 200 µl of washed and buffer equilibrated anti-Flag magnetic beads ( Sigma-Aldrich, St. Louis, MO ) for 2 hr at 4°C. The beads were washed once with lysis buffer and three times with washing buffer (20 mM Tris/HCl pH 7.5; 0.5 mM EDTA; 0.1% tween; 10% glycerol; 150 mM NaCl; 5 mM MgCl2; 0.5 mM PMSF; 1X Complete Mini EDTA-Free (Roche, Switzerland); 1X PhosSTOP (Roche)). Proteins were eluted with 5 bed volume of elution buffer (20 mM Tris/HCl pH 8; 0.5 mM EDTA; 0.1% tween; 10% glycerol; 150 mM NaCl; 5 mM MgCl2; 0.5 mM PMSF; 1X Complete Mini EDTA-Free (Roche); 1X PhosSTOP (Roche); 100 µg/mL Flag peptide) for 1 hr at 4°C. Proteins were separated by SDS-PAGE, stained with colloidal blue, and 7 bands covering the entire lane were excised for each sample. In-gel digestion was performed overnight by using trypsin (Gold, Promega, Madison, WI). Peptides extracted from each band were analyzed by nanoLC-MS/MS using an Ultimate 3000 system (Dionex, Thermo Scientific, Waltham, MA ) coupled to a LTQ-Orbitrap XL mass spectrometer (Thermo Scientific). Raw spectra were processed using Mascot through Proteome Discoverer (version 1.4, Thermo Scientific) and further analyzed in *my*ProMS (*Poullet et al., 2007*). Data was searched

against a Swissprot fasta database containing *S. cerevisiae* sequences. Only proteins found in two experiments and not in the control IPs were considered candidates.

## Co-immunoprecipitation

$6.10^8$ cells were harvested, washed one time with PBS, and lyzed in 1.5 ml lysis buffer (20 mM HEPES/KOH pH7.5; 150 mM NaCl; 0.5% Triton X-100; 10% Glycerol; 1 mM MgCl2; 2 mM EDTA; 1 mM PMSF; 1X Complete Mini EDTA-Free (Roche); 1X PhosSTOP (Roche); 125 U/mL benzonase (Sigma)) and glass beads three times for 30 s in a Fastprep instrument (MP Biomedicals, Santa Ana, CA). The lysate was cleared by centrifugation at 13,000 g for 5 min. 25 µl of Protein G magnetic beads (New England Biolabs, Ipswich, MA) (equilibrated 1:1 with lysis buffer) and primary antibodies (5 µg of mouse monoclonal anti-FLAG antibody M2 (Sigma), 3 µg of c-Myc monoclonal antibody 9E10 (Santa Cruz, Dallas, TX) or 5 µg of HA monoclonal antibody 16B12 (Covance, Princeton, NJ)) were added. The tubes were incubated overnight at 4°C. The magnetic beads were washed four times with 1 mL of wash buffer (20 mM HEPES/KOH pH7.5; 150 mM NaCl; 0.5% Triton X-100; 5% Glycerol; 1 mM MgCl2; 2 mM EDTA; 1 mM PMSF; 1X Complete Mini EDTA-Free (Roche); 1X Phos-STOP (Roche)) and resuspended in 30 µl of 2xSDS protein sample buffer. The beads were heated at 95°C for 10 min and loaded in duplicate or triplicate onto a 4–12% SDS-polyacrylamide gel. The proteins were then blotted to PVDF and probed for Flag, HA or Myc-tagged protein with corresponding antibodies. Signal was detected using the SuperSignal West Pico Chemiluminescent Substrate (ThermoFisher).

## Chromatin immunoprecipitation and real-time quantitative PCR

For each meiotic time point, $2.10^8$ cells were processed as described (*Borde et al., 2009*), with the following modifications: lysis was performed in Lysis buffer plus 1 mM PMSF, 50 µg/mL Aprotinin and 1X Complete Mini EDTA-Free (Roche), using 0.5 mm zirconium/silica beads (Biospec Products, Bartlesville, OK). We used 2 µg of the mouse monoclonal anti-FLAG antibody M2 (Sigma) and 30 µL Protein G magnetic beads (New England Biolabs) or 1.6 µg of c-Myc monoclonal antibody (9E10, Santa Cruz) and 50 µL PanMouse IgG magnetic beads (Thermo Scientific). Quantitative PCR was performed from the immunoprecipitated DNA or the whole-cell extract using a 7900HT Fast Real-Time PCR System (Applied Biosystems, Thermo Scientific) and SYBR Green PCR master mix (Applied Biosystems) as described (*Borde et al., 2009*). Results were expressed as % of DNA in the total input present in the immunoprecipitated sample and normalized to the negative control site in the middle of *NFT1*, a 3.5 kb long gene. Primers for *GAT1*, *BUD23*, *HIS4LEU2* and *NFT1* have been described (*Brachet et al., 2015*; *Sommermeyer et al., 2013*). The *ERG1* primers had the following sequence: Forward, 5' GCAACACATGGACCGATAACAC 3' and reverse, 5' GCCGACAACACAA TTACCTACGA 3'.

## Protein purification and biochemical assays

The *MLH2* gene was amplified from SK1 and cloned into pFB-MBP-MLH3-His (*Ranjha et al., 2014*) to make pFB-MBP-MLH2-his by following the same cloning procedure as was used in the preparation of pFB-MBP-MLH3-His (*Ranjha et al., 2014*). To prepare pFB-8xHis_MLH1, we amplified a region in the promoter area before GST sequence from pFB-GST-MLH1 plasmid (*Ranjha et al., 2014*) using forward primer pFB_his_F and reverse primer pFB_his_R. The reverse primer carried 8x histidine gene sequence. The amplified product was cloned into pFB-GST-MLH1 using *Eco*RV and *Bam*HI restriction enzymes. This introduced 8xhis tag before the MLH1 gene creating pFB-8xHis-MLH1. The *MER3* cDNA was amplified from SK1 and cloned into pFB-MBP-MLH3-His, replacing MLH3 gene to create pFB-MBP-MER3-His plasmid. *mer3R893E* and *mer3K167A* mutations were introduced into the pFB-MBP-MER3-His plasmid by site-directed mutagenesis. The His-Mlh1 and MBP-Mlh2 proteins were co-expressed as a heterodimer in *Sf9* insect cells and purified similarly to Mlh1-Mlh3 (*Ranjha et al., 2014*). The protein obtained was around 1 mg from 1 L culture that resulted in a concentration of 6 µM. Mer3, Mer3R893E and Mer3K167A proteins were also expressed and purified using the same method, and the total yield of purified proteins were ~0.2 mg from 2.4 L culture and the protein concentrations were 1.5 µM each. RPA was purified as described before (*Kantake et al., 2003*).

Structures such as ssDNA, dsDNA, Y- Structure, HJ, Nicked HJ and Open HJ were prepared as described previously (*Ranjha et al., 2014*). D-loop I, D-loop II and bubbled DNA were prepared essentially as described before with details of minor modifications available on request (*Opresko et al., 2004*).

The binding assays for Mer3 was carried out in 15 µl volume in Tris acetate, pH 7.5, 1 mM DTT, 100 µg/ml BSA, 1.3 ng/µl poly(dI-dC) (competitor DNA, 20 fold molar excess in nucleotides) and DNA substrate (0.5 nM, in molecules). The reactions also contained either 2 mM magnesium ($+Mg^{2+}$) or 3 mM EDTA ($-Mg^{2+}$). The reactions were assembled on ice and incubated at 30°C for 30 min. At the end of incubation, 5 µl of binding dye (50% glycerol and 0.25% bromophenol blue) was added to each sample. The products were separated on native 4% polyacrylamide gels (acrylamide: bisacrylamide 19:1) at 4°C. For Mlh1-Mlh2 binding, the reactions were done in the same manner as Mer3 but without competitor DNA and in the presence of 2 mM magnesium ($+Mg^{2+}$). After separation, the gels were dried and exposed to storage phosphor screens (GE Healthcare). The screens were scanned using Typhoon FLA 9500 (GE Healthcare) and quantified by Image Quant software.

Helicase assays were conducted as described before (*Nakagawa and Kolodner, 2002b*). 1 nM DNA substrate (in molecules) was used in all experiments. Reactions were assembled on ice, initiated by adding Mer3 and incubated for 30 min at 30°C. The reactions were then stopped with 5 µl of stop solution containing 150 mM EDTA, 2% SDS, 30% glycerol, 0.25% bromophenol blue and 1 µl Proteinase K (14–22 mg/ml, Roche) for 15 min at 30°C. The products were separated by electrophoresis on 10% polyacrylamide gel (acrylamide: bisacrylamide 19:1, Biorad) at room temperature. Gels were dried and quantified as above.

The nuclease assays were performed as described before (*Ranjha et al., 2014*). The products were analyzed on 1% agarose gels, post-stained with Gel Red and imaged on a UV illuminator.

For the protein interaction assays, MBP-Mlh2-Mlh1, MBP-Mlh2 or GST-Mlh1 were expressed in S*f*9 cells. The cells were lysed with lysis buffer (50 mM Tris-HCl, pH 7.5, 325 mM NaCl, 1 mM DTT, 1 mM EDTA, 30 µg/ml leupeptine, 1 mM phenylmethylsolfonyl fluoride and 1:400 (v/v) Sigma protease inhibitory mix [P8340]) and MBP-Mlh2-Mlh1 or MBP-Mlh2 proteins were immobilized on amylose resin (New England Biolabs), or GST-Mlh1 was immobilized on gluthatione resin (Qiagen, Germany). The resin was washed with wash buffer (50 mM Tris-HCl, pH 7.5, 80 mM NaCl (or as indicated), 0.2% NP40 detergent, 2 mM EDTA and 1:1000 (v/v) Sigma protease inhibitor mix [P8340]) and incubated for 60 min at 4°C with 2.5 µg of recombinant Mer3 protein. The resin was washed five times batchwise with wash buffer and Mer3 was eluted with 100 µl wash buffer containing 20 mM maltose (Sigma-Aldrich) in amylose pulldown, or 20 mM gluthatione (Amresco, Solon, OH) in GST pulldown. The MBP or GST tag was then cleaved by addition of prescission protease. The samples were separated by 10% SDS-PAGE and visualized by silver staining or Western blot (anti-His; A00186-100; GenScript; 1:2500 and anti-MBP; E8038S; New England Biolabs; 1:8000).

## Genome-wide meiotic recombination events inferred from octad analysis

Genomic DNA from the eight meiotic products of single meioses was prepared using Qiagen Genomic-tip 100 kit, and sequenced using an Illumina HiSeq 2500 instruments. We studied four wild-type octads, two *mlh2Δ* meioses, two complete *mer3R893E* octads, one *mer3R893E* octad where the sequence of only seven cells was retrieved. We used the two complete octads for tracts lengths and interference, and the three octads for mean CO numbers. For *mer3K167A* and *mer3167A mlh2Δ*, we studied two octads of each genotype. To infer recombination events from octad sequences, we first established a list of 74,911 SNPs by aligning the S288C chromosomes to the SK1 chromosomes using LAGAN (*Brudno et al., 2003*). Indels were not considered, and all positions located in LTR, retrotransposons and telomeres were discarded. We used the sequences from SGD for S288C (S288C_reference_genome_R64-1-1_20110203) and the sequence made by Scott Keeney's lab (http://cbio.mskcc.org/public/SK1_MvO/) for SK1. Next, sequencing reads from octad were aligned on the S288C and SK1 reference sequences using the BWA software (*Li and Durbin, 2009*) to genotype SNPs and deduce recombination events. Only perfectly matching reads to the reference sequences were taken into account. The average sequencing depth was 70X. The SNP genotype was considered as valid when there was one type of reads only (S288C or SK1) and if the number of reads was comprised between 15 and 150, or when there was two types of reads if the most abundant was comprised between 40 and 150 and the less abundant was below 5. Deduction of strand

transfers events were performed largely as in (*Martini et al., 2011*). Events that were separated by less than 1.5 kb were considered to have arisen from the same DSB and were therefore combined.

The strength of crossover interference was estimated from the distribution of inter-CO distances by maximum-likelihood fitting of gamma distribution (using fitdistr from R MASS package). The shape parameter γ of the best fitted gamma distribution is indicative of the strength of CO interference (γ = 1: no interference, γ > 1: positive interference, higher γ: stronger interference) (*Anderson et al., 2011*).

Raw sequence data have been deposited in the NCBI Sequence Read Archive (http://www.ncbi.nlm.nih.gov/Traces/sra/) under the accession number: SRP075437.

## Source data

Numerical values underlying all graphs are contained in *Supplementary file 4*

## Acknowledgements

We thank Arnaud De Muyt for critical reading of the manuscript, Bruce Fetcher, Mathilde Grelon and Deborah Bourc'his for reagents, and Nancy Hollingsworth for advice. We thank the NGS and Bioinformatics platforms of the Institut Curie.

## Additional information

### Funding

| Funder | Grant reference number | Author |
|---|---|---|
| Agence Nationale de la Recherche | ANR-11-LBX-0044 | Valérie Borde |
| Agence Nationale de la Recherche | ANR-12-BSV6-0009 | Valérie Borde |
| Agence Nationale de la Recherche | ANR-13-BSV6-0012-01 | Bertrand Llorente |
| Agence Nationale de la Recherche | ANR-15-CE11-0011 | Jean-Baptiste Charbonnier Petr Cejka Valérie Borde |
| Schweizerischer Nationalfonds zur Förderung der Wissenschaftlichen Forschung | PP00P3 133636 | Petr Cejka |
| Fondation pour la Recherche Médicale | DEP20131128517 | Valérie Borde |
| Ligue Contre le Cancer | postdoctoral fellowship | Khan Md Muntaz |
| French Infrastructure for Integrated Structural Biology | ANR-10-INSB-05-01 | Raphaël Guérois Jean-Baptiste Charbonnier |

The funders had no role in study design, data collection and interpretation, or the decision to submit the work for publication.

### Author contributions

YD, RK, Designed and performed research and analyzed data, Revised the article; LR, Designed, performed and analyzed all in vitro biochemical experiments; CA, KMM, Performed research and analyzed data; RG, Modeled and designed Mer3 mutation; M-CM-K, RL, Analyzed octad sequencing data; FD, DL, Carried out MS and proteomic analyses; BL, Supervised and analyzed octad sequencing data; J-BC, Provided key structural information and participated in the design of the study; PC, Designed and supervised experiments with purified proteins; VB, Designed and supervised the overall research, Wrote the manuscript with input from all the authors

### Author ORCIDs

Petr Cejka, http://orcid.org/0000-0002-9087-032X
Valérie Borde, http://orcid.org/0000-0001-6520-2461

## Additional files

### Supplementary files

• Suplementary file 1. Spore viabilities.

• Supplementary file 2. Genotypes of strains used in this study.

• Supplementary file 3. Strains used in each Figure panel.

• Supplementary file 4. Primary data for graphs in each figure.

### Major datasets

The following dataset was generated:

| Author(s) | Year | Dataset title | Dataset URL | Database, license, and accessibility information |
|---|---|---|---|---|
| Borde V | 2016 | WGS of S. cerevisiae hybridmsh2delta otherwise WT or mutant octad cells | https://trace.ncbi.nlm.nih.gov/Traces/sra/sra.cgi?study=SRP075437 | Publicly available at the NCBI Sequence Read Archive (accession no. SRP075437) |

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
