## [Decision Letter]

Thank you for submitting your article "Concerted action of the MutLβ heterodimer and Mer3 helicase regulates the global extent of meiotic gene conversion" for consideration by *eLife*. Your article has been reviewed by three peer reviewers, including Hannah L Klein (Reviewer #3), and the evaluation has been overseen by a Reviewing Editor and Kevin Struhl as the Senior Editor.

The reviewers have discussed the reviews with one another and the Reviewing Editor has drafted this decision to help you prepare a revised submission.

Summary:

The manuscript provides a novel function for the Mlh1-Mlh2 (MutLβ) heterodimer, a conserved protein complex whose primary function has, to date, remained elusive. It shows that Mlh1-Mlh2 interacts with Mer3, a meiosis-specific helicase previously implicated as having crossover-promoting function, and that the two work together to limit the length of heteroduplex tracts formed during meiotic recombination in a manner that appears to be largely independent of Mer3 helicase activity. Because all proteins in this study are conserved in many eukaryotes, including mammals, it is highly likely that these findings will be of general applicability.

Essential revisions:

A reorganization and better introduction of the key facts are needed, the paper would benefit from having all in vitro interaction data presented in one place so that direct comparisons can be made or model recombination substrate interaction data should be presented independent of meiotic data.

In addition, you should consider having someone not immediately in the field read the manuscript to make sure that the big picture and the flow of the data are easy to understand. The following suggestions should also be attended in a revised manuscript.

Understanding the larger aspects of the paper are difficult for the non-specialist. A cartoon figure of meiotic recombination and segregation at the beginning would help. For example, in the first two paragraphs of the Introduction, it is not easy to follow how recombination shuffles parental genomes through COs and NCOs or how these concepts extend to allelic shuffling, transmission distortion or gene conversion tract length, few lines later.

Interaction between Mer3 and Mlh1 or Mlh2 is well-documented, but not with the intact Mlh1-Mlh2 complex, even though purification of this complex is documented. Demonstrating this interaction should be the primary in vitro interaction data presented. While the paper presents the Mer3-MutLβ interaction as being mediated through Mlh2, it is clear from the in vitro data (Figure 2, Figure 3) that Mlh1 is also capable of interacting with Mer3, although with lower affinity. This needs to be more even-handedly discussed in the paper text.

Yeast two-hybrid data will be better interpreted if they are not scattered throughout different supplementary figures and the standards to define an interaction are the same throughout the manuscript. For example, in Figure 2—figure supplement 1A, limited growth on -His plates and none on -His -Ade is taken as a signal of a significant interaction, while later (in Figure 4—figure supplement 1) the relevant assay seems to have shifted towards whether or not growth occurs on -His -Ade.

There's much more focus on 2-hybrid assays using Mlh1 and Mer3 than on Mlh2 and Mer3, even though authors argue that Mlh2 is responsible for bridging between Mlh1 and Mer3.

All two-hybrid data should be presented in a single place and two-hybrid experiments should be presented in parallel with internal standards (for example, including Mlh2 + Mlh1 on every plate).

Analysis of Mlh1-Mer3 and Mlh2-Mer3 interactions would be greatly strengthened if experiments were done in mlh2- and mlh1- mutants, to directly test bridging interactions.

The model intermediate-binding activity of MutLβ is quite weak, as compared to Mer3, and only poorly discriminates model recombination intermediates from dsDNA. It's hard to directly assess this, since the critical data for MutLβ (at 10nM) is obscured by the plot in Figure 4. This can be solved doing the following: 1) present graphs of MutLβ and Mer3 binding together in one figure, along with the cartoons of the model substrates, but move the actual gels to a supporting figure; 2) use either a log or split-X axis, so that data for 10nM can be clearly discerned; 3) don't assign Kd values to the interaction between MutLβ and the various substrates-the current data are just too sparse-instead describe binding in relative terms.

Data values in Figure 4 are discordant with those in panel D; the numerical values for Mer3R893E in panel C are similar to those for wild-type Mer3 in panel D. These data sets need to be reconciled, and data for *mer3R893E* with a mismatch at HIS4-LEU2 should be included in panel D. One possible explanation for this discordance is the use of the NFT1 locus, which is not expected to bind Mer3 or MutLβ, as "normalization" standard, which means that all values are "normalized" to a background value that might be vulnerable to experimental variation. What happens if un-normalized (i.e. just Mlh2-ChIP/input) is used?

Data presentation for meiotic phenotypes are somewhat disorganized and confusing; given that mer3-hd is analyzed in Figure 6, it would make sense to present spore viability data for this mutant along with the other spore viability data.

Data regarding partial suppression of the spore viability defect of *zmm* mutants, while interesting, is of limited effect and possible mechanisms are not clear; moreover, a compelling case for the phenotypic equivalence of *mlh2Δ* and *mer3R893E* is made on the basis of tetrad data alone, and the msh4 suppression data could easily be omitted without compromising the conclusions.

Analysis of gene conversion and crossing over in SK1-S288c hybrids is quite compelling, but the presentation needs to include data on the mean number of noncrossovers per meiosis, and also comparison with γ values for true *zmm* mutants from other studies.

---

## [Author Response]

*Essential revisions:*

*A reorganization and better introduction of the key facts are needed, the paper would benefit from having all in vitro interaction data presented in one place so that direct comparisons can be made or model recombination substrate interaction data should be presented independent of meiotic data.*

Following this suggestion, we have reorganized the Results section, so that now the in vitro interaction data are presented in a separate figure, Figure 4.

In addition, DNA binding assays of recombinant Mer3 and MutLB are now grouped, together with the MutLβ endonuclease assay in Figure 6.

*In addition, you should consider having someone not immediately in the field read the manuscript to make sure that the big picture and the flow of the data are easy to understand.*

We had the manuscript read by a cell biologist colleague and added explanatory sequences at several places in the manuscript, especially in the Introduction.

*The following suggestions should also be attended in a revised manuscript.*

*Understanding the larger aspects of the paper are difficult for the non-specialist. A cartoon figure of meiotic recombination and segregation at the beginning would help. For example, in the first two paragraphs of the Introduction, it is not easy to follow how recombination shuffles parental genomes through COs and NCOs or how these concepts extend to allelic shuffling, transmission distortion or gene conversion tract length, few lines later.*

As suggested, we have added an introductory Figure 1 depicting how meiotic recombination shuffles alleles, and how it can result in gene conversion.

*Interaction between Mer3 and Mlh1 or Mlh2 is well-documented, but not with the intact Mlh1-Mlh2 complex, even though purification of this complex is documented. Demonstrating this interaction should be the primary* in vitro *interaction data presented.*

The reviewer may have missed it, but the interaction between purified Mer3 and Mlh1-Mlh2 complex was in Figure 3. It is now on Figure 4, and we present it in a clearer way in the text.

*While the paper presents the Mer3-MutLβ interaction as being mediated through Mlh2, it is clear from the* in vitro *data (Figure 2, Figure 3) that Mlh1 is also capable of interacting with Mer3, although with lower affinity. This needs to be more even-handedly discussed in the paper text.*

We have added additional evidence that Mer3 interacts directly with Mlh1 by showing that the 2 hybrid interaction between Mer3 and Mlh1 still occurs in the absence of Mlh2. This and the evolutionary conservation of this interaction is now more extensively discussed (see response to reviewer comment below).

*Yeast two-hybrid data will be better interpreted if they are not scattered throughout different supplementary figures and the standards to define an interaction are the same throughout the manuscript. For example, in Figure 2—figure supplement 1A, limited growth on -His plates and none on -His -Ade is taken as a signal of a significant interaction, while later (in Figure 4—figure supplement 1) the relevant assay seems to have shifted towards whether or not growth occurs on -His -Ade.*

*There's much more focus on 2-hybrid assays using Mlh1 and Mer3 than on Mlh2 and Mer3, even though authors argue that Mlh2 is responsible for bridging between Mlh1 and Mer3.*

*All two-hybrid data should be presented in a single place and two-hybrid experiments should be presented in parallel with internal standards (for example, including Mlh2 + Mlh1 on every plate).*

We group our answer to the last three comments. We have added in Materials and methods a sentence defining “interaction”: "interaction is defined compared to the growth seen in the negative control consisting of the combination between the GAL4BD-bait protein in the presence of GAL4AD-only (“empty” on the figures). Any combination that grows better than this control on the selective media is considered as an interaction".

In addition, to compare individual experiments, we have added the interaction between Mlh1 and Mlh3 as an internal standard in each experiment. This is seen in Figure 4—figure supplement 2 and Figure 4—figure supplement 3.

*Analysis of Mlh1-Mer3 and Mlh2-Mer3 interactions would be greatly strengthened if experiments were done in mlh2- and mlh1- mutants, to directly test bridging interactions.*

This is a great suggestion, and indeed we have tested two hybrid interaction between Mlh1 and Mer3 in a *mlh2∆* mutant, and two hybrid interaction between Mlh1 and Mer3R893, defective in the interaction with Mlh2. We still see an interaction between Mlh1 and Mer3, meaning that the two hybrid interaction is not mediated by Mlh2, in agreement with the direct interaction between recombinant proteins.

*The model intermediate-binding activity of MutLβ is quite weak, as compared to Mer3, and only poorly discriminates model recombination intermediates from dsDNA. It's hard to directly assess this, since the critical data for MutLβ (at 10nM) is obscured by the plot in Figure 4. This can be solved doing the following: 1) present graphs of MutLβ and Mer3 binding together in one figure, along with the cartoons of the model substrates, but move the actual gels to a supporting figure.*

As suggested, we have moved the Mer3 binding data to the main manuscript, and this data is presented next to MutLβ DNA binding for direct comparison (Figure 6).

*2) Use either a log or split-X axis, so that data for 10nM can be clearly discerned.*

We have changed the scale of the X axis to a log scale, which makes all concentrations easy to distinguish (Figure 6).

*3) Don't assign Kd values to the interaction between MutLβ and the various substrates-the current data are just too sparse-instead describe binding in relative terms.*

We no longer mention Kd values, but just binding in relative terms in the text as suggested (subsection “Mlh1-Mlh2 preferentially recognizes D-loops and related branched DNA structures and lacks endonuclease activity”)

*Data values in Figure 4 are discordant with those in panel D; the numerical values for Mer3R893E in panel C are similar to those for wild-type Mer3 in panel D. These data sets need to be reconciled, and data for mer3R893E with a mismatch at HIS4-LEU2 should be included in panel D. One possible explanation for this discordance is the use of the NFT1 locus, which is not expected to bind Mer3 or MutLβ, as "normalization" standard, which means that all values are "normalized" to a background value that might be vulnerable to experimental variation. What happens if un-normalized (i.e. just Mlh2-ChIP/input) is used?*

The experiments done in Figure 5 (previously Figure 4) were done in parallel (2 independent sets of WT and mutant) with the same lot of antibody, whereas the set of experiments presented in Figure 5 were done later, with a different lot of antibody. We believe this is the cause of the discrepancy between the two panels. As suggested, we have added Mlh2 ChIP data in the *mer3R893E* strain that has a mismatch at HIS4LEU2 in panel 5F. This allows reproducing a similar decrease in Mlh2 binding compared to *MER3*, as the one seen in panels 5C and 5E.

*Data presentation for meiotic phenotypes are somewhat disorganized and confusing; given that mer3-hd is analyzed in Figure 6, it would make sense to present spore viability data for this mutant along with the other spore viability data.*

We thank the reviewer for this suggestion. We have moved all the meiotic progression and spore viability data of the mer3-hd mutant to the current Figure 7 (previously Figure 5).

*Data regarding partial suppression of the spore viability defect of zmm mutants, while interesting, is of limited effect and possible mechanisms are not clear; moreover, a compelling case for the phenotypic equivalence of mlh2Δ and mer3-R893E is made on the basis of tetrad data alone, and the msh4 suppression data could easily be omitted without compromising the conclusions.*

The phenotypic equivalence between *mlh2Δ* and *mer3R893E* is also based on the similar delay in meiotic divisions, as well as the increase of spore viability of *zmm* mutants. We felt that the *msh4* data reinforce the findings seen for *zip4,* another *zmm* mutant, and we decided to keep both sets of data, which also reinforce the observation that *mlh2∆* and *mer3R893E* phenotypes are identical.

*Analysis of gene conversion and crossing over in SK1-S288c hybrids is quite compelling, but the presentation needs to include data on the mean number of noncrossovers per meiosis, and also comparison with γ values for true zmm mutants from other studies.*

We have added, in Figure 8—figure supplement 1, the mean number of NCO in the analyzed meioses. In the text (subsection “Mlh2 limits the extent of sequences involved in meiotic recombination intermediates through its interaction with Mer3”, first paragraph) we compare the γ values of our data to those obtained for WT and several *zmm* mutants in a previous study that used the same approach (Chen et al., 2008).